# Robust Semi-Supervised Metric Learning Meets with High Dimensionality

## Abstract

Classical semi-supervised metric learning usually formulates the objectives via maximizing/minimizing the ratio formed with must-links and cannot-links. However, the presence of noise and adversarial attacks can result in incorrect pairings, which will diminish the reliability of learned projection directions. To develop a robust distance metric learning method, we propose a new objective for distance metric learning using the $\ell_{2,q}$-norm ($0 < q < 2$) distances which will alleviate the influence of outliers or adversarial attacks. We develop an algorithm that will decrease the objective monotonically with updates. Additionally, we address computational burdens (e.g., $\mathcal{O}(d^3)$ complexity, where $d$ is the size of features) by introducing a 2D metric learning algorithm and extending it to arbitrary dimensions with kernel methods, backed by theoretical guarantees. Extensive empirical evaluations consistently demonstrate the superiority of our methods across various experimental setups.

## 1 Introduction

Most clustering/classification algorithms rely on defining a distance metric to assess the similarity between instances (Kulis et al., 2013; Kaya & Bilge, 2019; Bellet et al., 2013; Yang & Jin, 2006). Applying an appropriate distance metric that can capture important features from instances is critical to improving performance, as illustrated in Figure 2. While some general metrics are available, they often treat all features equally, which is inadequate as certain features hold more significance than others. Therefore, how to learn a distance metric that efficiently captures the idiosyncrasies of data with good quality has emerged as a prevalent research focus (Cakir et al., 2019; López-Sánchez et al., 2019; Karlinsky et al., 2019). Traditional techniques require explicit class labels and labeled data for classification transformations (Goldberger et al., 2004). However, obtaining precise labels in real-world scenarios is costly and time-consuming, prompting the development of semi-supervised methods that learn distance metrics with limited supervisory information (Hoi et al., 2006). These methods typically assume either a small portion of labeled data or pairwise constraints between examples. Clearly, the latter type is weaker. Therefore extensive research has been done to learn distance metrics with pairwise relevance information: must-links and cannot-links (Xiang et al., 2008). For more comprehensive survey of various metric learning algorithms, we refer our readers to the classical monograph (Kulis et al., 2013) and references therein.

The goal of this paper is to develop a robust distance metric incorporating pairwise relevance relationships. Existing metric learning methods, often based on squared Frobenius norm distances, are susceptible to outliers, features, and adversarial attacks. To address this issue, we propose a robust distance metric learning objective using the $\ell_{2,q}$-norm, offering robustness against outliers/attacks for any $0 < q < 2$. In addition, existing methods typically vectorize images before optimizing the projection matrix $\mathbf{W}$ via eigenvalue decomposition, which becomes computationally demanding for large dimensions, such as $100 \times 100$ grayscale images ($d = 10000$). Additionally, vectorization distorts image structure, leading to reduced recognition accuracy. Inspired by above, we propose a 2D metric learning algorithm that avoids image vectorization, utilizing covariance matrices with dimensions $r \times r$, where $r = \min\{m, n\}$ for each $m \times n$ input image. This approach offers significant computational savings compared to existing methods, as shown in Figure 1. Besides, inspired by Kernel Principal Component Analysis (Schölkopf et al., 1997) and Kernel Support Vector Machine (Amari & Wu, 1999), we introduce a kernel-based metric learning algorithm to

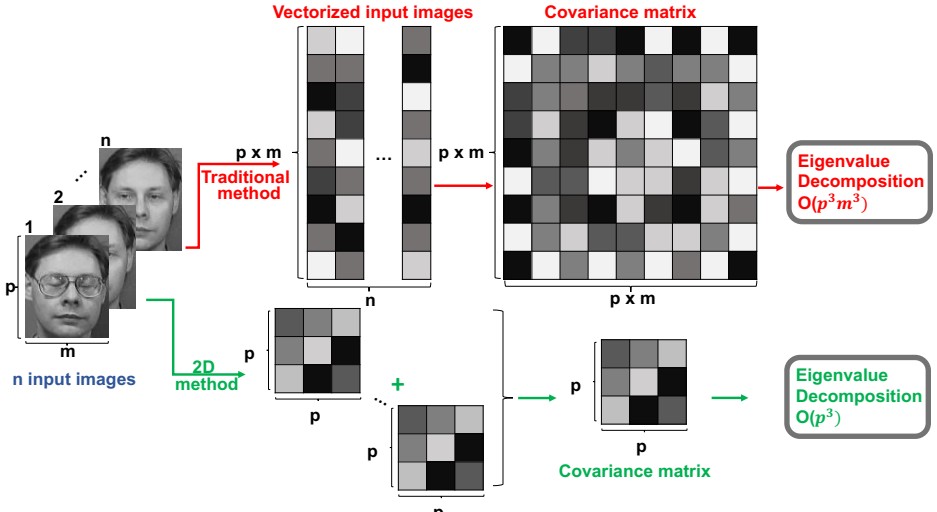

Figure 1: Upper flow denotes metric learning via traditional methods by vectorizing images, lower flow is the 2D metric learning method proposed in this paper, the covariance matrix size is significantly reduced.

address non-linearity in higher-dimensional spaces using kernel trick. This method ensures that the projection matrix $\mathbf{W}$ satisfies orthonormality constraints, even when obtained implicitly or in infinite dimensions. Our algorithms do not require explicit class labels; instead, they utilize pairwise relevance relationships in a semi-supervised manner, and they are designed for minimal computational costs and rapid convergence.

In Section 2, we will formulate the objective for metric learning and discuss its connection with Fisher's LDA before introducing a general framework for solving maximizing trace ratio problem. In Section 3, we discuss a more robust formulation against noise and adversarial samples which widely exist in real world. Section 4 depicts 2D metric learning which can be naturally extended to higher dimension and Section 5 generalizes into kernel version for metric learning. Section 6 entails convergence analysis, which concludes our method enjoys superlinear convergence rate and experiments in Section 7 demonstrate the superiority of our proposed methods over existing counterparts. Our main contributions are summarized as below:

- Propose a robust metric learning formulation and solve it with an efficient algorithm.

- Design novel 2D metric learning along with its robust version which works fast on high dimensional data, and propose an efficient algorithm which has superlinear convergence rate.

- Discuss and solve kernel version metric learning which goes beyond the linearity assumption.

## 2 Related Work

### 2.1 Mahalanobis Distance

Assume that we have a set of $n$ data points $\mathcal{X} = \{\mathbf{x}_i \in \mathbb{R}^p\}_{i=1}^n$ and two sets of pairwise constraints over the data points $\mathcal{X}$ are given under certain application context (Xing et al., 2003; Liu et al., 2019)[1]

$$\begin{cases} \mathcal{S} = \{(\mathbf{x}_i, \mathbf{x}_j) \mid \mathbf{x}_i \text{ and } \mathbf{x}_j \text{ are in the same class}\}; \\ \mathcal{D} = \{(\mathbf{x}_i, \mathbf{x}_j) \mid \mathbf{x}_i \text{ and } \mathbf{x}_j \text{ from different classes}\}, \end{cases} \tag{1}$$

He is happy     He is angry
She is happy     She is angry
He is very happy     He is very angry
She is very happy     She is very angry

Figure 2: An illustration to demonstrate the importance of metric learning. If we do $K$-means directly based on the word-count matrix generated from each sentence, then it may yield poor results. A good metric learning should be able to learn different weights for various features. For example, if 'happy' and 'angry' are assigned with significantly different weights, then $K$-means can work well on the learned metrics.

---

[1] We note that our model can be easily extended to *triplet relationship* $\mathcal{R} = \{(\mathbf{x}_i, \mathbf{x}_j, \mathbf{x}_l) \mid \mathbf{x}_i \text{ is more similar to } \mathbf{x}_j \text{ than to } \mathbf{x}_l\}$ where $\mathbf{x}_i - \mathbf{x}_l$ and $\mathbf{x}_i - \mathbf{x}_j$ can be conveyed in the numerator and denominator in Eq. (3) respectively.

where we denote $\mathcal{S}$ as must-links and $\mathcal{D}$ as cannot-links. Note that it is not necessary for all the data points in $\mathcal{X}$ to be involved in $\mathcal{S}$ or $\mathcal{D}$.

Given any two data points $\mathbf{x}_i$ and $\mathbf{x}_j$, the Mahalanobis distance between them is defined as:

$$\|\mathbf{x}_i - \mathbf{x}_j\|_{\mathbf{M}} = \sqrt{(\mathbf{x}_i - \mathbf{x}_j)^T \mathbf{M} (\mathbf{x}_i - \mathbf{x}_j)} \ , \tag{2}$$

where $\mathbf{M} \in \mathbb{R}^{p \times p}$ is the Mahalanobis distance metric (Shen et al., 2010; De Maesschalck et al., 2000), a symmetric matrix of size $p \times p$. In general, $\mathbf{M}$ is a valid metric if and only if $\mathbf{M}$ is a positive semi-definite matrix by satisfying the non-negativity and the triangle inequality conditions, *i.e.*, $\mathbf{M} \succeq 0$. The goal of robust metric learning is to learn an optimal square matrix $\mathbf{M}$ from a collection of data points $\mathcal{X}$ in the presence of outliers or adversarial attacks, coupled with a set of similar pairwise constraints $\mathcal{S}$ and a set of dissimilar pairwise constraints $\mathcal{D}$, such that the distances between the data point pairs in $\mathcal{S}$ are as small as possible, whilst those in $\mathcal{D}$ are as large as possible.

## 2.2 Metric Learning

Because $\mathbf{M}$ is positive semi-definite, we can reasonably write $\mathbf{M} = \mathbf{W}\mathbf{W}^T$, where $\mathbf{W} \in \mathbb{R}^{p \times r}$ with $r \leq p$. Thus the Mahalanobis distance under the metric $\mathbf{M}$ can be computed as $\|\mathbf{x}_i - \mathbf{x}_j\|_{\mathbf{M}} = \sqrt{(\mathbf{x}_i - \mathbf{x}_j)^T \mathbf{W}\mathbf{W}^T (\mathbf{x}_i - \mathbf{x}_j)} = \left\|\mathbf{W}^T (\mathbf{x}_i - \mathbf{x}_j)\right\|_2$, which defines a transformation $\mathbf{y} = \mathbf{W}^T \mathbf{x}$ under projection matrix $\mathbf{W}$. Our intuition is data points from different classes after projection are far away while from same class should be close, therefore we can formulate the objective as:

$$\max_{\mathbf{W}^T\mathbf{W}=\mathbf{I}} \frac{\sum\limits_{(\mathbf{x}_i,\mathbf{x}_j)\in\mathcal{D}} \left\|\mathbf{W}^T(\mathbf{x}_i-\mathbf{x}_j)\right\|_2^2}{\sum\limits_{(\mathbf{x}_i,\mathbf{x}_j)\in\mathcal{S}} \left\|\mathbf{W}^T(\mathbf{x}_i-\mathbf{x}_j)\right\|_2^2} = \frac{\sum\limits_{i=1}^{d} \left\|\mathbf{W}^T\mathbf{b}_i\right\|_2^2}{\sum\limits_{i=1}^{s} \left\|\mathbf{W}^T\mathbf{a}_i\right\|_2^2} = \frac{\left\|\mathbf{W}^T\mathbf{B}\right\|_F^2}{\left\|\mathbf{W}^T\mathbf{A}\right\|_F^2} = \frac{\text{tr}\left(\mathbf{W}^T\mathbf{S}_b\mathbf{W}\right)}{\text{tr}\left(\mathbf{W}^T\mathbf{S}_w\mathbf{W}\right)}, \tag{3}$$

where $\mathbf{A} = [\mathbf{a}_1, \mathbf{a}_2, \ldots, \mathbf{a}_s] \in \mathbb{R}^{p \times s}$ such that each column of $\mathbf{A}$ : $(\mathbf{x}_i - \mathbf{x}_j)$ satisfies $(\mathbf{x}_i, \mathbf{x}_j) \in \mathcal{S}$, and similarly $\mathbf{B} = [\mathbf{b}_1, \mathbf{b}_2, \ldots, \mathbf{b}_d] \in \mathbb{R}^{p \times d}$ such that each column of $\mathbf{B}$ : $(\mathbf{x}_i - \mathbf{x}_j)$ satisfies $(\mathbf{x}_i, \mathbf{x}_j) \in \mathcal{D}$. $\mathbf{S}_w = \sum_{(\mathbf{x}_i,\mathbf{x}_j)\in\mathcal{S}} (\mathbf{x}_i - \mathbf{x}_j)(\mathbf{x}_i - \mathbf{x}_j)^T$ is the covariance matrix of must-links and $\mathbf{S}_b = \sum_{(\mathbf{x}_i,\mathbf{x}_j)\in\mathcal{D}} (\mathbf{x}_i - \mathbf{x}_j)(\mathbf{x}_i - \mathbf{x}_j)^T$ denotes covariance of cannot-links. For the sake of brevity, we denote $|\mathcal{S}| = s$ and $|\mathcal{D}| = d$. In the above objective, the numerator term measures the scatteredness of different classes, while the denominator term denotes the compactness of the same class. Orthogonality constraint is to prevent degenerate solution(Xiang et al., 2008).

## 2.3 Connection with Fisher's Linear Discriminant Analysis

It is easy to notice that the above equation has the same objective as Fisher's Linear Discriminant Analysis where the only difference is the existence of orthonormality constraint on $\mathbf{W}$ in our formulation. By observing the independence of each column of $\mathbf{W}$, we can reformulate Eq. (3) as :

$$\max_{\mathbf{w}_i} \sum_{i=1}^{r} \mathbf{w}_i^T \mathbf{S}_b \mathbf{w}_i, \ s.t. \ \mathbf{w}_i^T \mathbf{S}_w \mathbf{w}_i = 1, \ \forall i \in [r]. \tag{4}$$

By making use of Lagrangian Multipliers, we obtain $\mathbf{S}_b \mathbf{w}_i = \lambda_i \mathbf{S}_w \mathbf{w}_i$ and $\max_{\mathbf{w}_i} \sum_{i=1}^{r} \mathbf{w}_i^T \mathbf{S}_b \mathbf{w}_i = \max \sum_{i=1}^{r} \lambda_i$. If we assume $\mathbf{S}_w$ is invertible and the projection is along different directions, then to obtain optimal $\mathbf{W}$, it is equivalent to solve $\mathbf{S}_w^{-1} \mathbf{S}_b \mathbf{w}_i = \lambda_i \mathbf{w}_i$ and find the top $r$ eigenvalues of $\mathbf{S}_w^{-1} \mathbf{S}_b$ and the corresponding eigenvectors. However, the main drawback of Fisher's LDA is as $\mathbf{S}_w^{-1} \mathbf{S}_b$ is not necessarily symmetric, each column of optimal $\mathbf{W}$ is generally non-orthogonal to each other, which means the new coordinate system formed by $\mathbf{W}$, has non-orthogonal axes. The reason to prefer orthogonal coordinates instead of general curvilinear coordinates is simplicity: many complications arise when coordinates are not orthogonal. Our experiments in Figure 4 demonstrate that the orthonormal constraint is nontrivial as it can not only obtain better objective, but also the classification accuracy is higher as it can avoid ill-conditioned coordinate base. We refer our readers to Daubechies (1993); Ninness et al. (1999) and references therein.

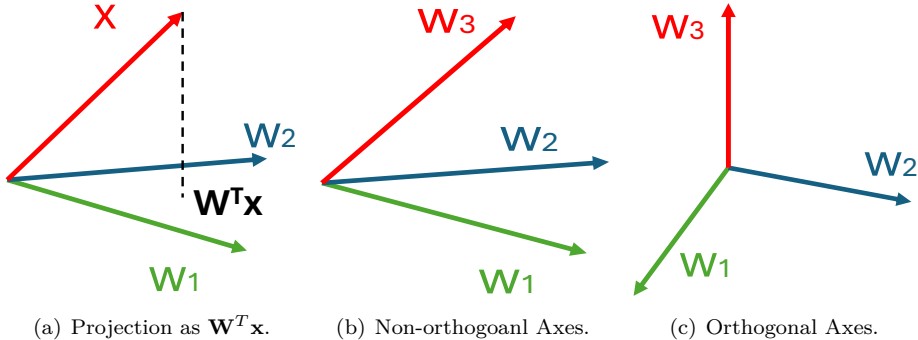

(a) Projection as $\mathbf{W}^T\mathbf{x}$.     (b) Non-orthogoanl Axes.     (c) Orthogonal Axes.

Figure 3: When $\|\mathbf{w}_i\|_2 = 1$, $\mathbf{W}^T\mathbf{x}$ is the projection in new axes system formed by columns of $\mathbf{W}$. Specifically, $\mathbf{w}_i^T\mathbf{x}$ determines the position in $i$-th axis. In Fisher's LDA formulation, each $\mathbf{w}_i$ is not orthogonal to others. In contrast, the orthonormality constraint ensures the new coordinate follows Cartesian system.

---

**Algorithm 1** The algorithm to solve the problem (5).

---

**Initialization:** $v \in \mathcal{C}$
**repeat**
    Calculate $\lambda = \frac{f(v)}{g(v)}$
    Update $v$ by solving the following problem:

$$v = \arg\max_{v \in \mathcal{C}} f(v) - \lambda g(v) \tag{6}$$

    **until** convergence

---

### 2.4 An Optimization Framework

The objective formulation above is inherently nonconvex due to the nonconvex nature of orthogonality constraint imposed on $\mathbf{W}$. In response to this challenge, we propose to address the problem effectively with an iterative algorithm. We first introduce a framework for maximization problem (Wang et al., 2014):

$$\max_{v \in \mathcal{C}} \frac{f(v)}{g(v)}, \quad \text{where } g(v) > 0 \ (\forall \ v \in \mathcal{C}). \tag{5}$$

The optimization procedure is described in Algorithm 1. The set $\mathcal{C}$ in our metric learning problem is Stiefel manifold $St(p, r)$ defined as $\{\mathbf{W} : \mathbf{W} \in \mathbb{R}^{p \times r}, \mathbf{W}^T\mathbf{W} = \mathbf{I}_r\}$. Several theorems are now in order.

**Theorem 2.1.** *By updating as Algorithm 1, the objective in (5) is monotonically non-decreasing.*

*Proof.* By definition $v^+ = \arg\max_{v \in \mathcal{C}} f(v) - \lambda g(v)$, we have: $f(v^+) - \lambda g(v^+) \geq f(v) - \lambda g(v) = 0$. Since $g(v) \geq 0$, therefore $f(v^+) - \lambda g(v^+) \geq 0 \implies \lambda^+ = \frac{f(v^+)}{g(v^+)} \geq \frac{f(v)}{g(v)} = \lambda$. $\square$

**Theorem 2.2.** *If the updated $v$ in Algorithm 1 is a stationary point of problem (6), the converged solution in Algorithm 1 is a stationary point of problem (5).*

*Proof.* Suppose the converged solution in Algorithm 1 is $v^*$. If $v^*$ is a stationary point of problem (6),

$$f'(v^*) - \frac{f(v^*)}{g(v^*)}g'(v^*) = 0 \Rightarrow f'(v^*)g(v^*) = f(v^*)g'(v^*). \tag{7}$$

On the other hand, if $v^*$ is a critical point of problem (5), then

$$(\frac{f(v)}{g(v)})'|_{v=v^*} = 0 \Rightarrow \frac{f'(v^*)g(v^*) - f(v^*)g'(v^*)}{g^2(v^*)} = 0. \tag{8}$$

Apparently, Eq. (7) is equivalent to (8) given the fact that $g(v) > 0$, which completes the proof. $\square$

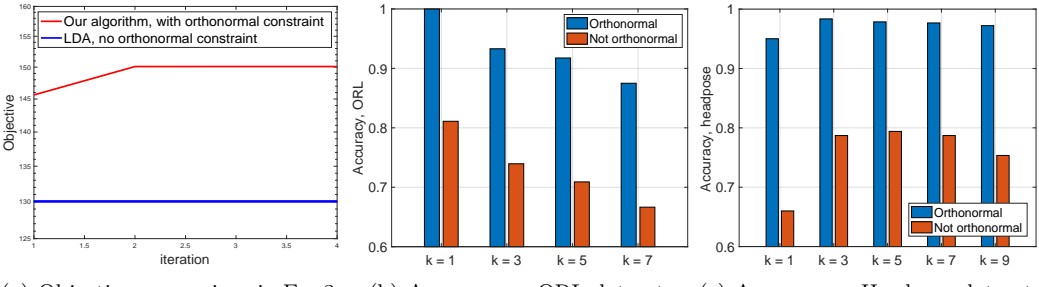

(a) Objective comparison in Eq. 3.  (b) Accuracy on ORL dataset.  (c) Accuracy on Headpose dataset.

Figure 4: Experiments show that the orthonormality constraint can not only obtain better objective (Fig. 4(a)), but also improve classification results on real-world datasets (Fig. 4(b)–Fig. 4(c)).

# 3 Robust Metric Learning

## 3.1 Problem Formulation

Eq. (3) quantifies the ratio between squared $\ell_2$-norm distances of pairs within must-links and cannot-links. Like other least square minimization models in machine learning and statistics, it is sensitive to outliers. Recent progress (Baccini et al., 1996; Gao, 2008; Ke & Kanade, 2004; Ding et al., 2006; Kwak, 2008; Wright et al., 2009) has shown that the $\ell_1$-norm or $\ell_{2,1}$-norm distance can promote robustness against outlier samples or features, which have been widely applied to replace the squared $\ell_2$-norm distance in many traditional machine learning methods, such as $\ell_{2,q}$-norm PCA (Wang et al., 2018). Inspired by the methods described above, we propose a general robust metric learning formulation based on Eq. (3) as:

$$\max_{\mathbf{W}^T\mathbf{W}=\mathbf{I}} \frac{\sum_{(\mathbf{x}_i,\mathbf{x}_j)\in\mathcal{S}} \left\|\mathbf{W}^T(\mathbf{x}_i-\mathbf{x}_j)\right\|_2^q}{\sum_{(\mathbf{x}_i,\mathbf{x}_j)\in\mathcal{D}} \left\|\mathbf{W}^T(\mathbf{x}_i-\mathbf{x}_j)\right\|_2^q} = \frac{\sum_{i=1}^s \left\|\mathbf{W}^T\mathbf{a}_i\right\|_2^q}{\sum_{i=1}^d \left\|\mathbf{W}^T\mathbf{b}_i\right\|_2^q} = \frac{\|\mathbf{A}^T\mathbf{W}\|_{2,q}}{\|\mathbf{B}^T\mathbf{W}\|_{2,q}} \;, \tag{9}$$

where $0 < q < 2$ [2] and $\|\mathbf{Z}\|_{2,q} = \sum_i \|\mathbf{Z}(i,:)\|_2^q$.

The above objective is obviously more challenging than Eq. 3 given the fact $q \neq 2$. As we will discuss later, we will propose a more generalized algorithm in which vanilla metric learning where $q = 2$ is a special case.

## 3.2 Algorithm to Solve Eq. (6)

In this subsection, we are to develop an algorithm to obtain the optimal solution of $\mathbf{W}$ in the following problem:

$$\max_{\mathbf{W}^T\mathbf{W}=\mathbf{I}} \|\mathbf{A}^T\mathbf{W}\|_{2,q} - \lambda\|\mathbf{B}^T\mathbf{W}\|_{2,q}, \tag{10}$$

which is equivalent to:

$$\min_{\mathbf{W}^T\mathbf{W}=\mathbf{I}} \lambda\|\mathbf{B}^T\mathbf{W}\|_{2,q} - \|\mathbf{A}^T\mathbf{W}\|_{2,q}. \tag{11}$$

We start with a lemma that will be the foundation for analysis:

**Lemma 3.1.** $\nabla_{\mathbf{W}}\|\mathbf{A}^T\mathbf{W}\|_{2,1} = \mathbf{A}\mathbf{D}\mathbf{A}^T\mathbf{W}$, where $\mathbf{D}$ is a diagonal matrix with $\mathbf{D}(i,i) = \frac{1}{\|\mathbf{a}_i^T\mathbf{W}\|}$.

*Proof of Lemma 3.1.*

$$\nabla_{\mathbf{W}}\|\mathbf{A}^T\mathbf{W}\|_{2,1} = \sum_i \nabla_{\mathbf{W}}\|\mathbf{a}_i^T\mathbf{W}\| = \sum_i \mathbf{a}_i \frac{1}{\|\mathbf{a}_i^T\mathbf{W}\|}\mathbf{a}_i^T\mathbf{W} = \sum_i \mathbf{a}_i\mathbf{D}(i,i)\mathbf{a}_i^T\mathbf{W} = (\sum_i \mathbf{a}_i\mathbf{D}(i,i)\mathbf{a}_i^T)\mathbf{W} = \mathbf{A}\mathbf{D}\mathbf{A}^T\mathbf{W},$$

$$\tag{12}$$

---

[2]Note our algorithm also works for $q \geq 2$ or even $q \leq 0$, but that makes less sense as our goal is to make the model robust to outliers, therefore $q > 2$ will make it more sensitive to noise.

where the first equation comes from the definition of the $\ell_{2,q}$-norm for matrices, the second equation comes from the fact that $\nabla_{\mathbf{x}}\|\mathbf{x}\| = \frac{\mathbf{x}}{\|\mathbf{x}\|}$. $\qquad\square$

Based upon the lemma, we turn to study the $\ell_{2,q}$-norm case:

**Lemma 3.2.** $\nabla_{\mathbf{W}}\|\mathbf{A}^T\mathbf{W}\|_{2,q} = \mathbf{A}\mathbf{D}\mathbf{A}^T\mathbf{W}$, where $\mathbf{D}$ is a diagonal matrix with $\mathbf{D}(i,i) = q\|\mathbf{a}_i^T\mathbf{W}\|^{q-2}$.

*Proof of Lemma 3.2.*

$$
\begin{aligned}
\nabla_{\mathbf{W}}\|\mathbf{A}^T\mathbf{W}\|_{2,q} &= \sum_i \nabla_{\mathbf{W}}\|\mathbf{a}_i^T\mathbf{W}\|^q = \sum_i q\|\mathbf{a}_i^T\mathbf{W}\|^{q-1}\nabla_{\mathbf{W}}\|\mathbf{a}_i^T\mathbf{W}\| \\
&= \sum_i q\|\mathbf{a}_i^T\mathbf{W}\|^{q-1}\mathbf{a}_i\frac{1}{\|\mathbf{a}_i^T\mathbf{W}\|}\mathbf{a}_i^T\mathbf{W} = \sum_i \mathbf{a}_i\mathbf{D}(i,i)\mathbf{a}_i^T\mathbf{W} \\
&= (\sum_i \mathbf{a}_i\mathbf{D}(i,i)\mathbf{a}_i^T)\mathbf{W} = \mathbf{A}\mathbf{D}\mathbf{A}^T\mathbf{W},
\end{aligned}
\tag{13}
$$

where the first equation comes from the definition of matrix $\ell_{2,q}$-norm, the second equation comes from the fact that $\nabla_{\mathbf{x}}\mathbf{y}^q = q\mathbf{y}^{q-1}\nabla_{\mathbf{x}}\mathbf{y}$ and the second line is directly from Lemma 3.1. One can see if $q = 2$, then the gradient becomes $2\mathbf{A}\mathbf{A}^T\mathbf{W}$ which is in accordance with the vanilla squared Frobenius norm case. Therefore, our formulation is a more general case of traditional metric learning. $\qquad\square$

Denote $f(\mathbf{W}) = \lambda\|\mathbf{B}^T\mathbf{W}\|_{2,q} - \|\mathbf{A}^T\mathbf{W}\|_{2,q}$, then $\nabla f(\mathbf{W}) = \lambda\mathbf{B}\mathbf{D_B}\mathbf{B}^T\mathbf{W} - \mathbf{A}\mathbf{D_A}\mathbf{A}^T\mathbf{W}$.

### 3.3 Algorithm to Solve Eq. (11) with Retraction

Given the first order (gradient) information of the objective, it is not surprising to use projected gradient descent method where as long as we can find an appropriate stepsize, we can guarantee the objective will be monotonically decreasing. Meanwhile, the orthonormality constraint is nonconvex, which inspires us to propose a backtracking line search style algorithm. For sake of further analysis, we start with the following definition:

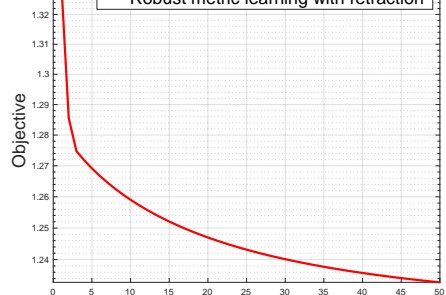

**Definition 3.3.** *A retraction on a differentiable manifold $\mathcal{C}$ is a smooth mapping Retr from the agent space of $\mathcal{C}$: $T\mathcal{C}$, onto $\mathcal{C}$ satisfying the following two conditions, $Retr_{\mathbf{X}}\mathcal{C}$ denotes the restriction of Retr onto $T_{\mathbf{X}}\mathcal{C}$: 1. $Retr_{\mathbf{X}}(0) = \mathbf{X}$, $\forall \mathbf{X} \in \mathcal{C}$; 2. For any $\mathbf{X} \in \mathcal{C}$ and $\delta \in T_{\mathbf{X}}\mathcal{C}$, it holds that $\lim_{\delta \to 0}\frac{\|Retr_{\mathbf{X}}(\delta) - (\mathbf{X}+\delta)\|_F}{\|\delta\|_F} = 0$.*

Figure 5: Objective update on synthetic data demonstrates our retraction algorithm can make the objective monotonically decreasing, which validates our theoretical analysis.

For the Stiefel manifold $St(p,r)$, common retractions include the polar decomposition $Retr_{\mathbf{X}}^{polar}(\delta) = (\mathbf{X}+\delta)(\mathbf{I}+\delta^T\delta)^{-1/2}$, the QR decomposition $Retr_{\mathbf{X}}^{QR}(\delta) = qf(\mathbf{X}+\delta)$, where $qf(\mathbf{X})$ is the Q factor of the QR factorization of $\mathbf{X}$. For a matrix $\mathbf{W} \in \mathbb{R}^{p\times r}$ with $r \leq p$, the total cost of computing the orthogonal projection is $8pr^2 + O(r^3)$ flops, while if $\mathbf{W} = \mathbf{X}+\delta$ and $\delta \in T_{\mathbf{X}}\mathcal{C}$ then the polar decomposition takes only $3pr^2 + O(r^3)$ flops and the QR decomposition takes only $2pr^2 + O(r^3)$. So if we can get a $\delta \in T_{\mathbf{X}}\mathcal{C}$ efficiently, we can utilize cheaper retraction operation rather than the expensive projection via singular value decomposition.

Based on the proximal gradient method described in Algorithm 2, in order to leverage retractions to handle the Stiefel manifold constraint, we need to find a descent direction in the tangent space $T_{\mathbf{W}_k}\mathcal{C}$, which is formulated as:

$$
\mathbf{V}_k = \arg\min_{\mathbf{V}} \langle\mathrm{grad}f(\mathbf{W}_k), \mathbf{V}\rangle + \frac{1}{2t}\|\mathbf{V}\|_F^2, \text{ s.t. } \mathbf{V} \in T_{\mathbf{W}_k}\mathcal{C}, \tag{14}
$$

where $\mathrm{grad}f$ denotes the Riemannian gradient of $f$. And using the fact that for $\mathbf{V} \in T_{\mathbf{W}_k}\mathcal{C}$ we have $\langle\mathrm{grad}f(\mathbf{W}_k), \mathbf{V}\rangle = \langle\nabla f(\mathbf{W}_k), \mathbf{V}\rangle$, we can simply solve the descent direction $\mathbf{V}$ without computing the

---

**Algorithm 2** Robust metric learning algorithm to solve Eq. (11) with retractions.

---

**Input:** $\epsilon, 0 < \gamma < 1$.
**Initialization:** $t$.
**repeat**
    Obtain $\mathbf{V}_k$ by Eq. (18)
    Set $\alpha = 1$
    While $f(Retr_{\mathbf{W}_k}(\alpha\mathbf{V}_k)) > f(\mathbf{W}_k) - \frac{\alpha\|\mathbf{V}_k\|_F^2}{2t}$
      $\alpha = \gamma\alpha$
    End While
    $\mathbf{W}_{k+1} = Retr_{\mathbf{W}_k}(\alpha\mathbf{V}_k)$
**until** satisfying stopping criterion $\|\mathbf{V}_k\|_F \le \frac{\epsilon}{L}$

---

Riemannian gradient $\mathrm{grad} f$:

$$\mathbf{V}_k = arg\,min\,l(\mathbf{V}) = arg\min_{\mathbf{V}}\langle\nabla f(\mathbf{W}_k), \mathbf{V}\rangle + \frac{1}{2t}\|\mathbf{V}\|_F^2, \text{ s.t. } \mathbf{V} \in T_{\mathbf{W}_k}\mathcal{C}. \tag{15}$$

With the definition of the tangent space to $\mathcal{C} = St(p, r)$ being $T_{\mathbf{W}}\mathcal{C} = \{\mathbf{V}|\mathbf{V}^T\mathbf{W} + \mathbf{W}^T\mathbf{V} = 0\}$, we can obtain $\mathbf{V}$ by checking the Karush–Kuhn–Tucker (KKT) conditions for the following Lagrangian function with a Lagrange multiplier $\Gamma$:

$$\min\mathcal{L}(\mathbf{V}, \Gamma) = \langle\nabla f(\mathbf{W}_k), \mathbf{V}\rangle + \frac{1}{2t}\|\mathbf{V}\|_F^2 - \langle\mathbf{V}^T\mathbf{W} + \mathbf{W}^T\mathbf{V}, \Gamma\rangle. \tag{16}$$

The KKT conditions are

$$0 \in \partial_{\mathbf{V}}\mathcal{L}(\mathbf{V}, \Gamma), \quad \mathbf{V}^T\mathbf{W} + \mathbf{W}^T\mathbf{V} = 0, \tag{17}$$

and we can obtain the following closed-form solution for $\mathbf{V}_k$ associated with $\mathbf{W}_k$:

$$\mathbf{V}_k = t\left(\frac{(\mathbf{W}_k\nabla f(\mathbf{W})^T\mathbf{W}_k + \mathbf{W}_k\mathbf{W}_k^T\nabla f(\mathbf{W}_k))}{2\|\mathbf{W}_k\|_F^2} - \nabla f(\mathbf{W}_k)\right). \tag{18}$$

We summarize the procedure by using retractions with a line search in Algorithm 2.

Here we also show the convergence properties of Algorithm 2, we present the following facts about retractions (Boumal et al., 2019) that will be leveraged in the later analysis:

**Lemma 3.4.** *Let $\mathcal{C}$ be a compact embedded submanifold of the Euclidean space, for all $\mathbf{X} \in \mathcal{C}$ and $\delta \in T_{\mathbf{X}}\mathcal{C}$, there exist constants $C_1 > 0$ and $C_2 > 0$ such that the following two inequalities hold:*

$$\|Retr_{\mathbf{X}}(\delta) - \mathbf{X}\|_F \le C_1\|\delta\|_F, \quad \|Retr_{\mathbf{X}}(\delta) - (\mathbf{X} + \delta)\|_F \le C_2\|\delta\|_F^2. \tag{19}$$

First, we want to show that we can get a descent direction $\mathbf{V}$ in $T_{\mathbf{W}_k}\mathcal{C}$ by solving Eq. (18).

**Lemma 3.5.** *With $l(\mathbf{V})$ being the objective in Eq. (15), for any $\alpha \in [0, 1]$, we have*

$$l(\alpha\mathbf{V}_k) - l(0) \le \frac{\alpha^2 - 2\alpha}{2t}\|\mathbf{V}_k\|_F^2. \tag{20}$$

*Proof.* $l$ is $\frac{1}{t}$-strongly convex, then

$$l(\mathbf{V}_1) \ge l(\mathbf{V}_2) + \langle\nabla l(\mathbf{V}_2), \mathbf{V}_1 - \mathbf{V}_2\rangle + \frac{1}{2}\|\mathbf{V}_1 - \mathbf{V}_2\|_F^2, \tag{21}$$

and when $\mathbf{V}_1, \mathbf{V}_2 \in T_{\mathbf{W}_k}\mathcal{C}$, we have $\langle\nabla l(\mathbf{V}_2), \mathbf{V}_1 - \mathbf{V}_2\rangle = \langle Proj_{T_{\mathbf{W}_k}\mathcal{C}}\nabla l(\mathbf{V}_2), \mathbf{V}_1 - \mathbf{V}_2\rangle$, and $0 \in Proj_{T_{\mathbf{W}_k}\mathcal{C}}\nabla l(\mathbf{V}_2)$ based on the optimality condition. Then, let $\mathbf{V}_1 = 0, \mathbf{V}_2 = \mathbf{V}_k$ in Eq. (21), we have

$$l(0) \ge l(\mathbf{V}_k) + \frac{1}{2t}\|\mathbf{V}_k\|_F^2 = \langle\nabla f(\mathbf{W}_k), \mathbf{V}_k\rangle + \frac{1}{t}\|\mathbf{V}_k\|_F^2, \tag{22}$$

therefore, we have $l(\alpha\mathbf{V}_k) - l(0) = \langle\nabla f(\mathbf{W}_k), \alpha\mathbf{V}_k\rangle + \frac{1}{2t}\|\alpha\mathbf{V}_k\|_F^2 \le \frac{\alpha^2-2\alpha}{2t}\|\mathbf{V}_k\|_F^2.$    □

We now show that the objective sequence $\{f(\mathbf{W}_k)\}$ is monotonically decreasing in Algorithm 2:

**Lemma 3.6.** *For any $t > 0$, there exists a constant $\bar{\alpha} > 0$ such that for any $0 < \alpha \leq \min\{1, \bar{\alpha}\}$, the inequality in the line search procedure, i.e. $f(Retr_{\mathbf{W}_k}(\alpha\mathbf{V}_k)) > f(\mathbf{W}_k) - \frac{\alpha\|\mathbf{V}_k\|_F^2}{2t}$, is satisfied, and the objective sequence $\{f(\mathbf{W}_k)\}$ generated by Algorithm 2 satisfies*

$$f(\mathbf{W}_{k+1}) - f(\mathbf{W}_k) \leq -\frac{\alpha\|\mathbf{V}_k\|_F^2}{2t}. \tag{23}$$

*Proof.* With Lemma 3.4, let $\mathbf{W}_k^+ = \mathbf{W}_k + \alpha\mathbf{V}_k$, then for any $\alpha > 0$ we have

$$f(Retr_{\mathbf{W}_k}(\alpha\mathbf{V}_k)) - f(\mathbf{W}_k) \leq \langle \nabla f(\mathbf{W}_k), Retr_{\mathbf{W}_k}(\alpha\mathbf{V}_k) - \mathbf{W}_k \rangle + \frac{L}{2}\|Retr_{\mathbf{W}_k}(\alpha\mathbf{V}_k) - \mathbf{W}_k\|_F^2$$
$$\leq C_2\|\nabla f(\mathbf{W}_k)\|_F\|\alpha\mathbf{V}_k\|_F^2 + \alpha\langle \nabla f(\mathbf{W}_k), \mathbf{V}_k \rangle + \frac{LC_1^2}{2}\|\alpha\mathbf{V}_k\|_F^2. \tag{24}$$

Since $\nabla f$ is continuous on $\mathcal{C}$, we can safely say $\|\nabla f(\mathbf{W})\|_F$ is upper bounded by a constant number $G > 0$, thus let $c_0 = C_2 G + \frac{LC_1^2}{2}$, we can get

$$\begin{aligned}
f(Retr_{\mathbf{W}_k}(\alpha\mathbf{V}_k)) - f(\mathbf{W}_k) &\leq \alpha\langle \nabla f(\mathbf{W}_k), \mathbf{V}_k \rangle + c_0\alpha^2\|\mathbf{V}_k\|_F^2 \\
&= c_0\alpha^2\|\mathbf{V}_k\|_F^2 + l(\alpha\mathbf{V}_k) - \frac{1}{2t}\|\alpha\mathbf{V}_k\|_F^2 - l(0) \\
&\leq c_0\alpha^2\|\mathbf{V}_k\|_F^2 - \frac{1}{2t}\|\alpha\mathbf{V}_k\|_F^2 + \frac{\alpha^2}{2t}\|\mathbf{V}_k\|_F^2 - \frac{\alpha}{t}\|\mathbf{V}_k\|_F^2 \\
&= (c_0 - \frac{1}{\alpha t})\|\alpha\mathbf{V}_k\|_F^2,
\end{aligned} \tag{25}$$

where $l(\mathbf{V})$ is defined in Eq. (15) and the second inequality comes from Lemma 3.5. By setting $\bar{\alpha} = \frac{1}{2tc_0}$, we guarantee that for $0 < \alpha \leq \min\{1, \bar{\alpha}\}$, $f(Retr_{\mathbf{W}_k}(\alpha\mathbf{V}_k)) - f(\mathbf{W}_k) \leq -\frac{\alpha}{2t}\|\mathbf{V}_k\|_F^2$. □

**Definition 3.7.** $\mathbf{W}_k$ *is an $\epsilon$-stationary point of $g(\mathbf{W})$ if $\|\mathbf{V}_k\|_F \leq \frac{\epsilon}{L}$.*

**Theorem 3.8.** *Algorithm 2 will return an $\epsilon$-stationary point in $O(\frac{1}{\epsilon^2})$ iterations.*

*Proof.* Suppose Algorithm 2 doesn't terminate until the $K_{th}$ iteration, which means

$$\|\mathbf{V}_k\|_F > \frac{\epsilon}{L}, \forall k = 0, 1, \dots, K - 1, \tag{26}$$

and let $\alpha_k$ denote the actual $\alpha$ in the $k_{th}$ iteration, and we have $\alpha_k \geq \gamma\bar{\alpha}$ from Lemma 3.6, we have

$$f(\mathbf{W}_0) - f^* \geq f(\mathbf{W}_0) - f(\mathbf{W}_K) \geq \sum_{k=0}^{K-1} \frac{\alpha_k}{2t}\|\mathbf{V}_k\|_F^2 = \frac{t}{2}\sum_{k=0}^{K-1}\alpha_k\|\frac{\mathbf{V}_k}{t}\|_F^2 > \frac{t\epsilon^2}{2}\sum_{k=0}^{K-1}\alpha_k \geq \frac{Kt\epsilon^2}{2}\gamma\bar{\alpha}, \tag{27}$$

which implies that the number of iterations needed to obtain an $\epsilon$-stationary point in Algorithm 2 is $O(\frac{1}{\epsilon^2})$. □

## 4 2D Metric Learning

For a 2D dataset $\mathcal{X} = \{\mathbf{x}_i \in \mathbb{R}^{p \times m}\}_{i=1}^n$ (e.g., grayscale images) and given relevance relationships between certain pairs, we group paired data into $\mathcal{S}$ or $\mathcal{D}$. Unlike conventional metric learning, our 2D metric learning algorithm operates directly on 2D matrices instead of 1D vectors, eliminating the need to transform image data (Zhang & Zhou, 2005; Li & Yuan, 2005). This allows us to construct image covariance matrices directly from the original matrices, resulting in significantly reduced covariance matrix sizes, as shown in Fig. 1.

### 4.1 Problem Formulation

Given any two 2D data points $\mathbf{x}_i$ and $\mathbf{x}_j$, its Mahalanobis distance between them can be naturally given by:

$$\|\mathbf{x}_i - \mathbf{x}_j\|_{\mathbf{M}}^2 = \mathbf{tr}\left\{(\mathbf{x}_i - \mathbf{x}_j)^T \mathbf{M} (\mathbf{x}_i - \mathbf{x}_j)\right\} = \|\mathbf{W}^T (\mathbf{x}_i - \mathbf{x}_j)\|_F^2, \tag{28}$$

where $\mathbf{M} = \mathbf{W}\mathbf{W}^T \in \mathbb{R}^{p \times p}$.

Similar to the practices in traditional metric learning methods, suppose we are given a set of paired data instances in $\mathcal{D}$, along with some paired samples from $\mathcal{S}$ defined in Eq. (1), without loss of generality, we denote $\{\mathbf{D}_1, \mathbf{D}_2, \ldots, \mathbf{D}_d\}$ where $\mathbf{D}_k = \mathbf{x}_i^k - \mathbf{x}_j^k \in \mathbb{R}^{p \times m}$ is the difference between paired samples taken from set $\mathcal{D}$. Similarly, we can denote $\{\mathbf{S}_1, \mathbf{S}_2, \ldots, \mathbf{S}_s\}$ where $\mathbf{S}_k = \mathbf{x}_i^k - \mathbf{x}_j^k$ from set $\mathcal{S}$. By following the idea of Fisher's LDA (Fisher, 1936), the projection matrix $\mathbf{W}$ will make the distance within the same class as small as possible while setting the distance between different classes as large as possible, therefore we can formulate the 2D metric learning objective as:

$$\max_{\mathbf{W}^T\mathbf{W} = \mathbf{I}} \frac{\sum\limits_{(\mathbf{x}_i, \mathbf{x}_j) \in \mathcal{D}} \left\|\mathbf{W}^T (\mathbf{x}_i - \mathbf{x}_j)\right\|_F^2}{\sum\limits_{(\mathbf{x}_i, \mathbf{x}_j) \in \mathcal{S}} \left\|\mathbf{W}^T (\mathbf{x}_i - \mathbf{x}_j)\right\|_F^2} = \frac{\sum\limits_{i=1}^{d} \left\|\mathbf{W}^T \mathbf{D}_i\right\|_F^2}{\sum\limits_{i=1}^{s} \left\|\mathbf{W}^T \mathbf{S}_i\right\|_F^2}, \tag{29}$$

It is obvious that the denominator in Eq. (29) is nonnegative, therefore we could use the general framework in Algorithm 1 to optimize $\mathbf{W}$ in Eq. (29) with $\mathcal{C}$ corresponding to the orthonormal constraint on $\mathbf{W}$.

Following Algorithm 1, now we turn to optimize $\max\limits_{\mathbf{W}^T\mathbf{W} = \mathbf{I}} f(\mathbf{W}) - \lambda g(\mathbf{W})$. By Eq. (29) we have:

$$f(\mathbf{W}) = \sum_{i=1}^{d} \left\|\mathbf{W}^T \mathbf{D}_i\right\|_F^2 = \sum_{i=1}^{d} \mathbf{tr}\left(\mathbf{W}^T \mathbf{D}_i \mathbf{D}_i^T \mathbf{W}\right) = \mathbf{tr}\left(\mathbf{W}^T \mathbf{S}_b \mathbf{W}\right), \tag{30}$$

where $\mathbf{S}_b = \mathbf{D}_1\mathbf{D}_1^T + \mathbf{D}_2\mathbf{D}_2^T + \cdots + \mathbf{D}_d\mathbf{D}_d^T$ denotes the covariance matrix of data pairs from different clusters. Similarly, we can get $g(\mathbf{W}) = \sum_{i=1}^{s} \left\|\mathbf{W}^T \mathbf{S}_i\right\|_F^2 = \sum_{i=1}^{s} \mathbf{tr}\left(\mathbf{W}^T \mathbf{S}_i \mathbf{S}_i^T \mathbf{W}\right) = \mathbf{tr}\left(\mathbf{W}^T \mathbf{S}_w \mathbf{W}\right)$, with $\mathbf{S}_w = \mathbf{S}_1\mathbf{S}_1^T + \mathbf{S}_2\mathbf{S}_2^T + \cdots + \mathbf{S}_s\mathbf{S}_s^T$ denotes the covariance matrix from the same clusters.

The optimization problem now is in the following form:

$$\max_{\mathbf{W}} \quad \mathbf{tr}\left(\mathbf{W}^T \mathbf{S}_b \mathbf{W}\right) - \lambda \mathbf{tr}\left(\mathbf{W}^T \mathbf{S}_w \mathbf{W}\right) = \mathbf{tr}\left\{\mathbf{W}^T (\mathbf{S}_b - \lambda\mathbf{S}_w)\mathbf{W}\right\}, \tag{31}$$

with constraint $\mathbf{W}^T\mathbf{W} = \mathbf{I}_{r \times r}$. Though the constraint is nonconvex, still there is a closed solution for $\mathbf{W}$ by noticing the fact that $\mathbf{S}_b - \lambda\mathbf{S}_w$ is symmetric, which can be obtained by doing eigenvalue decomposition to $\mathbf{S}_b - \lambda\mathbf{S}_w \in \mathbb{R}^{p \times p}$ and pick the $r$ eigenvectors corresponding to the largest $r$ eigenvalues. The constraint $\mathbf{W}^T\mathbf{W} = \mathbf{I}_{r \times r}$ is automatically satisfied due to the property of symmetric matrix eigenvalue decomposition.

---

**Algorithm 3** 2D metric learning algorithm.

---

**Input:** $\{\mathbf{D}_1, \ldots, \mathbf{D}_d\}$, $\{\mathbf{S}_1, \ldots, \mathbf{S}_s\}$, $\mathbf{S}_w$, $\mathbf{S}_b$ and $r$.
**Initialization: W**
**repeat**
    Calculate $\lambda = \frac{\mathbf{tr}(\mathbf{W}^T \mathbf{S}_b \mathbf{W})}{\mathbf{tr}(\mathbf{W}^T \mathbf{S}_w \mathbf{W})}$;
    $[\mathbf{U}, \mathbf{V}] = eig(\mathbf{S}_b - \lambda\mathbf{S}_w,' descent')$;
    $\mathbf{W}^+ = \mathbf{U}\mathbf{V}^T$;
**until** convergence

---

---

**Algorithm 4** Robust 2D metric learning algorithm.

---

**Input:** $\{\mathbf{D}_1, \ldots, \mathbf{D}_d\}$, $\{\mathbf{S}_1, \ldots, \mathbf{S}_s\}$ and $r$.
**Initialization: W**
**repeat**

 Calculate $\lambda = \frac{\mathbf{tr}(\mathbf{W}^T \mathbf{S}_b \mathbf{W})}{\mathbf{tr}(\mathbf{W}^T \mathbf{S}_w \mathbf{W})}$
 **repeat**
  Calculate $\mathbf{S}_b, \mathbf{S}_w$ according to Eq. (34);
  $[\mathbf{U}, \mathbf{V}] = eig(\mathbf{S}_b - \lambda \mathbf{S}_w, \text{'}descent'\text{)}$;
  $\mathbf{W}^+ = \mathbf{U}\mathbf{V}^T$;
 **until** convergence
**until** convergence

---

Regarding the analysis of complexity, for the sake of simplicity, we assume the image dimension to be $n \times n$. Traditional metric learning methods vectorize an image to a vector of size $n^2$, so the covariance matrix size is $n^2 \times n^2$. Generally speaking, the complexity of eigenvalue decomposition is $\mathcal{O}(p^3)$ given matrix size $p \times p$. Therefore, the time complexity of 2D metric learning proposed by this paper is $\mathcal{O}(n^3)$ while traditional is $\mathcal{O}(n^6)$, which is a huge improvement, especially when $n$ is considerably large. Assume there are $K$ loops to update $\lambda$ in Algorithm 1, then the whole complexity is $\mathcal{O}(K * n^3)$ since the most significant consumption in the algorithm comes from eigenvalue decomposition.

### 4.2 Robust 2D Metric Learning

The same robust strategy in Section 3 can be applied to 2D metric learning as well:

$$\max_{\mathbf{W}^T\mathbf{W}=\mathbf{I}} \frac{\sum\limits_{(\mathbf{x}_i, \mathbf{x}_j) \in \mathcal{D}} \left\| \mathbf{W}^T (\mathbf{x}_i - \mathbf{x}_j) \right\|_F^q}{\sum\limits_{(\mathbf{x}_i, \mathbf{x}_j) \in \mathcal{S}} \left\| \mathbf{W}^T (\mathbf{x}_i - \mathbf{x}_j) \right\|_F^q} = \frac{\sum\limits_{i=1}^{d} \left\| \mathbf{W}^T \mathbf{D}_i \right\|_F^q}{\sum\limits_{i=1}^{s} \left\| \mathbf{W}^T \mathbf{S}_i \right\|_F^q} \ , \tag{32}$$

where $0 < q < 2$. Following earlier analysis, we turn to optimize:

$$\max_{\mathbf{W}^T\mathbf{W}=\mathbf{I}} \sum_{i=1}^{d} \left\| \mathbf{W}^T \mathbf{D}_i \right\|_F^q - \lambda \sum_{i=1}^{s} \left\| \mathbf{W}^T \mathbf{S}_i \right\|_F^q . \tag{33}$$

The robust 2D metric learning problem can be addressed using a very similar approach to vanilla 2D. By denoting $f(\mathbf{W}) = \sum\limits_{i=1}^{d} \left\| \mathbf{W}^T \mathbf{D}_i \right\|_F^q - \lambda \sum\limits_{i=1}^{s} \left\| \mathbf{W}^T \mathbf{S}_i \right\|_F^q$, then $\nabla f(\mathbf{W}) = q(\mathbf{S}_b - \lambda \mathbf{S}_w)\mathbf{W}$, where we define:

$$\mathbf{S}_w = \frac{\mathbf{S}_1 \mathbf{S}_1^T}{\|\mathbf{W}^T \mathbf{S}_1\|_F^{2-q}} + \cdots + \frac{\mathbf{S}_s \mathbf{S}_s^T}{\|\mathbf{W}^T \mathbf{S}_s\|_F^{2-q}}, \mathbf{S}_b = \frac{\mathbf{D}_1 \mathbf{D}_1^T}{\|\mathbf{W}^T \mathbf{D}_1\|_F^{2-q}} + \cdots + \frac{\mathbf{D}_d \mathbf{D}_d^T}{\|\mathbf{W}^T \mathbf{D}_d\|_F^{2-q}}. \tag{34}$$

Algorithm 4 is slightly different from Algorithm 3 in terms an inner loop to ensure $\mathbf{W}$ and $\mathbf{S}_w, \mathbf{S}_b$ converge.

## 5 Kernel Version Metric Learning

While the preceding sections offer methodologies for 1D and 2D data, it's important to acknowledge that in real-world scenarios, a substantial volume of data exists in high-dimensional spaces. Rather than transforming the tensor data into 2D or 1D formats, we present a versatile approach to address such data by leveraging the Kernel trick, which has demonstrated substantial promising performance, particularly when the data in the original space ($\mathbb{R}^d$) may not be well separable but can be effectively separated by projecting it into a higher-dimensional space. ($\mathbb{R}^n$) via $\Phi(\mathbf{x}_i)$ where $\Phi : \mathbb{R}^d \to \mathbb{R}^n$ (Liu et al., 2008; Leslie et al., 2001; Patle & Chouhan, 2013; Ye et al., 2009; Cai et al., 2011). Assume we are given $\{\mathbf{D}_1, \mathbf{D}_2, \ldots, \mathbf{D}_d\}$, $\{\mathbf{S}_1, \mathbf{S}_2, \ldots, \mathbf{S}_s\}$,

where $\mathbf{D}_i, \mathbf{S}_i \in \mathbb{R}^p$. Different from existing methods, the kernel method bypasses the explicit computation of eigenvalues, while it provides the efficient calculation of eigenvalues through the kernel trick instead. To the best of our knowledge, kernel metric learning in the form of min-max ratio optimization has not been previously explored in the literature.

### 5.1 Problem Formulation

Same as before, we formulate the kernel version objective as:

$$\max_{\mathbf{W}^T\mathbf{W}=\mathbf{I}} \frac{\sum\limits_{(\mathbf{x}_i,\mathbf{x}_j)\in\mathcal{D}} \left\| \mathbf{W}^T\Phi\left(\mathbf{x}_i-\mathbf{x}_j\right)\right\|_2^2}{\sum\limits_{(\mathbf{x}_i,\mathbf{x}_j)\in\mathcal{S}} \left\| \mathbf{W}^T\Phi\left(\mathbf{x}_i-\mathbf{x}_j\right)\right\|_2^2} = \frac{\sum\limits_{i=1}^{d} \left\| \mathbf{W}^T\Phi(\mathbf{D}_i)\right\|_2^2}{\sum\limits_{i=1}^{s} \left\| \mathbf{W}^T\Phi(\mathbf{S}_i)\right\|_2^2} = \frac{\left\| \mathbf{W}^T\Phi(\mathbf{D})\right\|_{\mathrm{F}}^2}{\left\| \mathbf{W}^T\Phi(\mathbf{S})\right\|_{\mathrm{F}}^2}. \tag{35}$$

Similar to the steps in the 2D version, we can first initialize $\lambda$ followed by optimizing:

$$\max_{\mathbf{W}^T\mathbf{W}=\mathbf{I}} \mathbf{tr}\left(\mathbf{W}^T\Phi(\mathbf{D})\Phi^T(\mathbf{D})\mathbf{W}\right) - \lambda\,\mathbf{tr}\left(\mathbf{W}^T\Phi(\mathbf{S})\Phi^T(\mathbf{S})\mathbf{W}\right) = \mathbf{tr}\left\{\mathbf{W}^T(\Phi(\mathbf{D})\Phi^T(\mathbf{D}) - \lambda\Phi(\mathbf{S})\Phi^T(\mathbf{S}))\mathbf{W}\right\}. \tag{36}$$

By observing that $\Phi(\mathbf{D})\Phi^T(\mathbf{D}) - \lambda\Phi(\mathbf{S})\Phi^T(\mathbf{S})$ is symmetric, we can transfer it into finding the $r$ eigenvectors corresponds to the top $r$ largest eigenvalue of $\Phi(\mathbf{D})\Phi^T(\mathbf{D}) - \lambda\Phi(\mathbf{S})\Phi^T(\mathbf{S})$. If we denote an eigenvector as $v$, and its corresponding eigenvalue as $\theta$, we have:

$$(\Phi(\mathbf{D})\Phi^T(\mathbf{D}) - \lambda\Phi(\mathbf{S})\Phi^T(\mathbf{S}))v = \theta v$$

$$\Leftrightarrow (\sum_{i=1}^{d}\Phi(\mathbf{D}_i)\Phi^T(\mathbf{D}_i) - \lambda\sum_{i=1}^{s}\Phi(\mathbf{S}_i)\Phi^T(\mathbf{S}_i))v = \theta v$$

$$\Leftrightarrow \sum_{i=1}^{d}\Phi(\mathbf{D}_i)\underbrace{\langle\Phi^T(\mathbf{D}_i), v\rangle}_{scalar} - \lambda\sum_{i=1}^{s}\Phi(\mathbf{S}_i)\underbrace{\langle\Phi^T(\mathbf{S}_i), v\rangle}_{scalar} = \theta v, \tag{37}$$

we see that $v$ is a linear combination of $\Phi(\mathbf{D}_i)$ and $\Phi(\mathbf{S}_i)$, therefore we have:

$$v = [\Phi(\mathbf{D}), \Phi(\mathbf{S})]\underbrace{\begin{bmatrix}\alpha_d \\ \alpha_s\end{bmatrix}}_{\alpha}, \tag{38}$$

where $\alpha \in R^{d+s}$. Now plug Eq. (38) in Eq. (37) we have:

$$(\Phi(\mathbf{D})\Phi^T(\mathbf{D}) - \lambda\Phi(\mathbf{S})\Phi^T(\mathbf{S}))[\Phi(\mathbf{D}), \Phi(\mathbf{S})]\alpha = \theta[\Phi(\mathbf{D}), \Phi(\mathbf{S})]\alpha, \tag{39}$$

which is equivalent to:

$$[\Phi(\mathbf{D})\mathbf{K}_{DD} - \lambda\Phi(\mathbf{S})\mathbf{K}_{SD}, \Phi(\mathbf{D})\mathbf{K}_{DS} - \lambda\Phi(\mathbf{S})\mathbf{K}_{SS}]\alpha = \theta[\Phi(\mathbf{D}), \Phi(\mathbf{S})]\alpha. \tag{40}$$

By multiplying $[\Phi(\mathbf{D}), \Phi(\mathbf{S})]^T$ to both sides of the above equation:

$$\begin{bmatrix}\mathbf{K}_{DD}\mathbf{K}_{DD} - \lambda\mathbf{K}_{DS}\mathbf{K}_{SD}\mathbf{K}_{DD}\mathbf{K}_{DS} - \lambda\mathbf{K}_{DS}\mathbf{K}_{SS} \\ \mathbf{K}_{SD}\mathbf{K}_{DD} - \lambda\mathbf{K}_{SS}\mathbf{K}_{SD}\mathbf{K}_{SD}\mathbf{K}_{DS} - \lambda\mathbf{K}_{SS}\mathbf{K}_{SS}\end{bmatrix}\alpha = \theta\begin{bmatrix}\mathbf{K}_{DD}\mathbf{K}_{DS} \\ \mathbf{K}_{SD}\mathbf{K}_{SS}\end{bmatrix}\alpha, \tag{41}$$

therefore, $\theta$ and $\alpha$ are the eigenvalue and eigenvector of

$$\begin{bmatrix}\mathbf{K}_{DD} & \mathbf{K}_{DS} \\ \mathbf{K}_{SD} & \mathbf{K}_{SS}\end{bmatrix}^{-1}\begin{bmatrix}\mathbf{K}_{DD}\mathbf{K}_{DD} - \lambda\mathbf{K}_{DS}\mathbf{K}_{SD} & \mathbf{K}_{DD}\mathbf{K}_{DS} - \lambda\mathbf{K}_{DS}\mathbf{K}_{SS} \\ \mathbf{K}_{SD}\mathbf{K}_{DD} - \lambda\mathbf{K}_{SS}\mathbf{K}_{SD} & \mathbf{K}_{SD}\mathbf{K}_{DS} - \lambda\mathbf{K}_{SS}\mathbf{K}_{SS}\end{bmatrix}, \tag{42}$$

where $\mathbf{K}_{DD} = \Phi(\mathbf{D})^T\Phi(\mathbf{D})$, $\mathbf{K}_{DS} = \Phi(\mathbf{D})^T\Phi(\mathbf{S})$, $\mathbf{K}_{SD} = \Phi(\mathbf{S})^T\Phi(\mathbf{D})$, $\mathbf{K}_{SS} = \Phi(\mathbf{S})^T\Phi(\mathbf{S})$ [3].

---

[3]Clearly, $\begin{bmatrix}\mathbf{K}_{DD} & \mathbf{K}_{DS} \\ \mathbf{K}_{SD} & \mathbf{K}_{SS}\end{bmatrix}$ is a Kernel matrix, therefore it is SPD. To avoid the singular case, in practice, we can take its inversion as $\left\{\begin{bmatrix}\mathbf{K}_{DD} & \mathbf{K}_{DS} \\ \mathbf{K}_{SD} & \mathbf{K}_{SS}\end{bmatrix} + \epsilon\mathbf{I}\right\}^{-1}$, where $\epsilon$ is a very small positive scalar.

---

**Algorithm 5** Kernel metric learning algorithm.

---

**Input:** $\{\mathbf{D}_1, \ldots, \mathbf{D}_d\}$, $\{\mathbf{S}_1, \ldots, \mathbf{S}_s\}$, $r$ and calculate $\mathbf{K}_{DD}$, $\mathbf{K}_{DS}$, $\mathbf{K}_{SD}$ and $\mathbf{K}_{SS}$ accordingly.
**Initialization:** $\lambda$
**repeat**
    Obtain eigenvalue $\theta$ and eigenvector $\alpha$ via Eq. (41);
    Calculate $\lambda$ as Eq. (43);
**until** convergence

---

It is worth noting that we do the eigenvalue decomposition based on the above matrix with dimension $(s + d) \times (s + d)$ instead of high dimension $n$, and it's computationally efficient to get eigenvectors $\mathbf{G} = [\alpha_1, \ldots, \alpha_r]$ corresponding to the largest $r$ largest eigenvalues $\theta$. After we obtain $\mathbf{G} \in R^{(s+d) \times r}$ composed of the first $r$ eigenvectors, according to Eq. (38), we obtain the projection matrix $\mathbf{W}$ as $[\Phi(\mathbf{D}), \Phi(\mathbf{S})]\mathbf{G}$. And therefore:

$$
\lambda = \frac{\left\|\mathbf{W}^T \Phi(\mathbf{D})\right\|_{\mathrm{F}}^2}{\left\|\mathbf{W}^T \Phi(\mathbf{S})\right\|_{\mathrm{F}}^2} = \frac{\mathbf{tr}\left(\mathbf{G}^T [\Phi(\mathbf{D}), \Phi(\mathbf{S})]^T \Phi(\mathbf{D}) \Phi^T(\mathbf{D}) [\Phi(\mathbf{D}), \Phi(\mathbf{S})]\mathbf{G}\right)}{\mathbf{tr}\left(\mathbf{G}^T [\Phi(\mathbf{D}), \Phi(\mathbf{S})]^T \Phi(\mathbf{S}) \Phi^T(\mathbf{S}) [\Phi(\mathbf{D}), \Phi(\mathbf{S})]\mathbf{G}\right)} = \frac{\mathbf{tr}\left\{\mathbf{G}^T \begin{bmatrix} \mathbf{K}_{DD}\mathbf{K}_{DD} & \mathbf{K}_{DD}\mathbf{K}_{DS} \\ \mathbf{K}_{SD}\mathbf{K}_{DD} & \mathbf{K}_{SD}\mathbf{K}_{DS} \end{bmatrix} \mathbf{G}\right\}}{\mathbf{tr}\left\{\mathbf{G}^T \begin{bmatrix} \mathbf{K}_{DS}\mathbf{K}_{SD} & \mathbf{K}_{DS}\mathbf{K}_{SS} \\ \mathbf{K}_{SS}\mathbf{K}_{SD} & \mathbf{K}_{SS}\mathbf{K}_{SS} \end{bmatrix} \mathbf{G}\right\}}.
$$
(43)

### 5.2 Image Recognition Via Kernel Version

In image clustering or classification, the cluster/class assignment is based on Euclidean distance in the lower dimension by projection matrix $\mathbf{W}$. Also, for some kernels, say Gaussian kernel, $\mathbf{W}$ can not be explicitly obtained as it is an infinite dimension mapping. Therefore, we seek image recognition implicitly via the Kernel trick. Assume we have anchor data samples $\mathcal{A} = \{\mathbf{a}_j\}(j = 1, \ldots, h)$ and query data $\mathbf{a}_i$, the squared Euclidean distance in transformed space is $\|\mathbf{W}^T \Phi(\mathbf{Q}_j)\|_F^2$, where $\mathbf{Q}_j = \mathbf{a}_j - \mathbf{a}_i (j = 1, \ldots, h)$. Therefore:

$$
\|\mathbf{W}^T \Phi(\mathbf{Q}_j)\|_F^2 = \mathbf{tr}\left(\mathbf{G}^T [\Phi(\mathbf{D}), \Phi(\mathbf{S})]^T \Phi(\mathbf{Q}_j) \Phi^T(\mathbf{Q}_j) [\Phi(\mathbf{D}), \Phi(\mathbf{S})]\mathbf{G}\right) = \mathbf{tr}\left\{\mathbf{G}^T \begin{bmatrix} \mathbf{K}_{D\mathbf{Q}_j}\mathbf{K}_{\mathbf{Q}_j D} & \mathbf{K}_{D\mathbf{Q}_j}\mathbf{K}_{\mathbf{Q}_j S} \\ \mathbf{K}_{S\mathbf{Q}_j}\mathbf{K}_{\mathbf{Q}_j D} & \mathbf{K}_{S\mathbf{Q}_j}\mathbf{K}_{\mathbf{Q}_j S} \end{bmatrix} \mathbf{G}\right\}.
$$
(44)

The kernel version metric learning algorithm is summarized in Algorithm 5. We end this section by pointing out that different kernel options may result in various performances, such as Linear, polynomial, and RBF kernel, hyper-parameter needs tuning when necessary, but $\mathbf{K}(\mathbf{x}, \mathbf{y}) = \langle \Phi(\mathbf{x}), \Phi(\mathbf{y}) \rangle$ is computationally efficient via the Kernel trick. In the kernel version metric learning algorithm, the main consumption goes to inversion and eigenvalue decomposition both on $(s + d) \times (s + d)$, therefore the whole complexity is $\mathcal{O}(K * (s + d)^3)$.

## 6 Convergence Analysis

We give the convergence rate of Algorithm 3 and Algorithm 5 in the following theorem:

**Theorem 6.1.** *The convergence rate of Algorithm 3 and Algorithm 5 is superlinear.*

*Proof.* We start the analysis with the definition of superlinear convergence rate: $\lim_{k \to \infty} \frac{\|F^{k+1} - F*\|}{\|F^k - F*\|} = 0$, where we define

$$
F^* = \max_{\mathbf{W}^T\mathbf{W}=\mathbf{I}} \frac{\mathbf{tr}\left(\mathbf{W}^T \mathbf{S}_b \mathbf{W}\right)}{\mathbf{tr}\left(\mathbf{W}^T \mathbf{S}_w \mathbf{W}\right)} := \max_{\mathbf{W}^T\mathbf{W}=\mathbf{I}} \frac{f(\mathbf{W})}{g(\mathbf{W})} = \max_{\mathbf{W}^T\mathbf{W}=\mathbf{I}} F(\mathbf{W}),
$$
(45)

and we have

$$
\mathbf{W}^* \in \operatorname{argmax} f(\mathbf{W}) - F(\mathbf{W}^*) g(\mathbf{W}).
$$
(46)

As $\mathbf{W} \in \mathbb{R}^{p \times r}$, we conclude the leading eigenvalues of $\mathbf{S}_b - F^* \mathbf{S}_w$ denoted as $\lambda_1, \ldots, \lambda_r$ satisfy $\sum_{i=1}^{r} \lambda_i = 0$. Many previous works have focused on the convergence rate of the constrained maximum trace ratio problem, including global linear or local quadratic, we refer the readers to Ngo et al. (2012); Zhang et al. (2010; 2013)

and the references therein. Our analysis will utilize the global linear convergence result established by Zhang et al. (2010), based on which we will show it is indeed superlinear:

$$F(\mathbf{W}^*) - F(\mathbf{W}^{k+1}) \leq (1 - \frac{1}{\kappa(\mathbf{S}_w)})(F(\mathbf{W}^*) - F(\mathbf{W}^k)), \tag{47}$$

where $\kappa(\mathbf{S}_w) = \sum_{i=1}^r \lambda_i(\mathbf{S}_w)/\sum_{i=1}^r \lambda_{p-i+1}(\mathbf{S}_w) > 1$ *almost surely.*

For any symmetric matrix $\mathbf{C} \in \mathbb{R}^{p \times p}$ and $\mathbf{V} = (\mathbf{v}_1, \mathbf{v}_2, \ldots, \mathbf{v}_p)$ be its eigenvectors in decreasing order *w.r.t.* eigenvalues, then we have

$$\sum_{i=1}^r \lambda_i - \mathbf{tr}\left(\mathbf{W}^T\mathbf{C}\mathbf{W}\right) = \sum_{i=1}^r \lambda_i - \sum_{i=1}^p \lambda_i \|\mathbf{W}^T\mathbf{v}_i\|_2^2 = \sum_{i=1}^r \lambda_i(1 - \|\mathbf{W}^T\mathbf{v}_i\|_2^2) - \sum_{i=r+1}^p \lambda_i \|\mathbf{W}^T\mathbf{v}_i\|_2^2, \tag{48}$$

with the fact that

$$\|\mathbf{W}^T\mathbf{v}_i\|_2^2 = \mathbf{v}_i^T\mathbf{W}\mathbf{W}^T\mathbf{v}_i \leq \|\mathbf{W}\mathbf{W}^T\|_2 = \|\mathbf{W}^T\mathbf{W}\|_2 = 1, \tag{49}$$

we have

$$\begin{aligned}
\sum_{i=1}^r \lambda_i - \mathbf{tr}\left(\mathbf{W}^T\mathbf{C}\mathbf{W}\right) &\geq \sum_{i=1}^r \lambda_i(1 - \|\mathbf{W}^T\mathbf{v}_i\|_2^2) - \lambda_{r+1}\sum_{i=r+1}^p \|\mathbf{W}^T\mathbf{v}_i\|_2^2 \\
&\geq \lambda_r(r - \sum_{i=1}^r \|\mathbf{W}^T\mathbf{v}_i\|_2^2) - \lambda_{r+1}(r - \sum_{i=1}^r \|\mathbf{W}^T\mathbf{v}_i\|_2^2) \\
&= (\lambda_r - \lambda_{r+1})(r - \sum_{i=1}^r \|\mathbf{W}^T\mathbf{v}_i\|_2^2).
\end{aligned} \tag{50}$$

Denote $\bar{\mathbf{V}} = (\mathbf{v}_1, \mathbf{v}_2, \ldots, \mathbf{v}_r) \in \mathbb{R}^{p \times r}$, for any square rotation matrix $\mathbf{R} \in \mathbb{R}^{r \times r}$ such that $\mathbf{R}^T\mathbf{R} = \mathbf{R}\mathbf{R}^T = \mathbf{I}$, with $\sigma_i(\cdot)$ denoting the singular values, we have

$$\begin{aligned}
\min_{\mathbf{W}^*} \|\mathbf{W} - \mathbf{W}^*\|_F^2 &\leq \min_{\mathbf{R}} \|\mathbf{W} - \bar{\mathbf{V}}\mathbf{R}\|_F^2 = \min_{\mathbf{R}} \|\mathbf{W}\|_F^2 + \|\bar{\mathbf{V}}\mathbf{R}\|_F^2 - 2\,\mathbf{tr}\left(\mathbf{W}^T\bar{\mathbf{V}}\mathbf{R}\right) \\
&= \min_{\mathbf{R}} 2(r - \mathbf{tr}\left(\mathbf{R}^T\bar{\mathbf{V}}^T\mathbf{W}\right)) = 2(r - \sum_{i=1}^r \sigma_i(\bar{\mathbf{V}}^T\mathbf{W})) \\
&\leq 2(r - \sum_{i=1}^r \sigma_i^2(\bar{\mathbf{V}}^T\mathbf{W})) = 2(r - \|\bar{\mathbf{V}}^T\mathbf{W}\|_F^2) = 2(r - \sum_{i=1}^r \|\mathbf{W}^T\mathbf{v}_i\|_2^2).
\end{aligned} \tag{51}$$

Combine Eq. (50) and Eq. (51), we get

$$\sum_{i=1}^r \lambda_i - \mathbf{tr}\left(\mathbf{W}^T\mathbf{C}\mathbf{W}\right) \geq \frac{\lambda_r - \lambda_{r+1}}{2}\min_{\mathbf{W}^*} \|\mathbf{W} - \mathbf{W}^*\|_F^2. \tag{52}$$

Denote $P(\mathbf{W}) = f(\mathbf{W}) - F(\mathbf{W}^*)g(\mathbf{W}) = g(\mathbf{W})(F(\mathbf{W}) - F^*) = \mathbf{tr}\left(\mathbf{W}^T(\mathbf{S}_b - F^*\mathbf{S}_w)\mathbf{W}\right)$, now set $\mathbf{C} = \mathbf{S}_b - F^*\mathbf{S}_w$, recall that the leading eigenvalues of $\mathbf{S}_b - F^*\mathbf{S}_w$ satisfy $\sum_{i=1}^r \lambda_i = 0$, and invoke Eq. (52), we get

$$\frac{\lambda_r - \lambda_{r+1}}{2}\min_{\mathbf{W}^*} \|\mathbf{W} - \mathbf{W}^*\|_F^2 \leq -\mathbf{tr}\left(\mathbf{W}^T\mathbf{C}\mathbf{W}\right) = g(\mathbf{W})(F^* - F(\mathbf{W})). \tag{53}$$

With the definition of $g(\mathbf{W})$ in Eq. (45), we have

$$g(\mathbf{W}) = \mathbf{tr}\left(\mathbf{W}^T\mathbf{S}_w\mathbf{W}\right) \leq \sum_{i=1}^r \lambda_i(\mathbf{S}_w), \tag{54}$$

therefore, by plugging $\mathbf{W} = \mathbf{W}^k$, we can obtain

$$\min_{\mathbf{W}^*} \|\mathbf{W} - \mathbf{W}^*\|_F^2 \leq \frac{2\sum_{i=1}^r \lambda_i(\mathbf{S}_w)}{\lambda_r - \lambda_{r+1}}(F^* - F(\mathbf{W}^k)) \leq \frac{2\sum_{i=1}^r \lambda_i(\mathbf{S}_w)}{\lambda_r - \lambda_{r+1}}(F^* - F(\mathbf{W}^0))(1 - \frac{1}{\kappa(\mathbf{S}_w)})^k. \quad (55)$$

On the other side,

$$\begin{aligned}
|g(\mathbf{W}^*) - g(\mathbf{W}^k)| &= |\langle \mathbf{W}^* - \mathbf{W}^k, \nabla g(\theta\mathbf{W}^k + (1-\theta)\mathbf{W}^*)\rangle| = |\langle \mathbf{W}^* - \mathbf{W}^k, 2\mathbf{S}_w(\theta\mathbf{W}^k + (1-\theta)\mathbf{W}^*)\rangle| \\
&\leq 2\|\mathbf{W}^* - \mathbf{W}^k\|_F\|\mathbf{S}_w\|_2(\theta\|\mathbf{W}^k\|_F + (1-\theta)\|\mathbf{W}^*\|_F) = 2\sqrt{r}\lambda_1(\mathbf{S}_w)\|\mathbf{W}^* - \mathbf{W}^k\|_F.
\end{aligned} \quad (56)$$

Since $\mathbf{W}^{k+1} = \text{argmax}_{\mathbf{W}} f(\mathbf{W}) - F(\mathbf{W}^k)g(\mathbf{W}) = \text{argmax}_{\mathbf{W}} g(\mathbf{W})\frac{f(\mathbf{W})}{g(\mathbf{W})} - g(\mathbf{W})F(\mathbf{W}^k) = \text{argmax}_{\mathbf{W}} g(\mathbf{W})(F(\mathbf{W}) - F(\mathbf{W}^k))$, we have

$$g(\mathbf{W}^{k+1})(F(\mathbf{W}^{k+1}) - F(\mathbf{W}^k)) \geq g(\mathbf{W}^*)(F(\mathbf{W}^*) - F(\mathbf{W}^k)), \quad (57)$$

dividing both sides by $g(\mathbf{W}^{k+1})$ and add $F(\mathbf{W}^*) - F(\mathbf{W}^k)$, we obtain

$$\begin{aligned}
F(\mathbf{W}^*) - F(\mathbf{W}^{k+1}) &\leq \frac{g(\mathbf{W}^*)}{g(\mathbf{W}^{k+1})}(F(\mathbf{W}^k) - F(\mathbf{W}^*)) + F(\mathbf{W}^*) - F(\mathbf{W}^k) \\
&= \frac{g(\mathbf{W}^{k+1}) - g(\mathbf{W}^*)}{g(\mathbf{W}^{k+1})}(F(\mathbf{W}^*) - F(\mathbf{W}^k)).
\end{aligned} \quad (58)$$

Thus, based on Eq. (55), Eq. (56) and Eq. (58), we can obtain

$$\begin{aligned}
\frac{F^* - F(\mathbf{W}^{k+1})}{F^* - F(\mathbf{W}^k)} &\leq \frac{g(\mathbf{W}^{k+1}) - g(\mathbf{W}^*)}{g(\mathbf{W}^{k+1})} \leq \frac{2\sqrt{r}\lambda_1(\mathbf{S}_w)\|\mathbf{W}^* - \mathbf{W}^k\|_F}{\sum_{i=1}^r \lambda_{p-i+1}(\mathbf{S}_w)} \\
&\leq \frac{2\sqrt{r}\lambda_1(\mathbf{S}_w)}{\sum_{i=1}^r \lambda_{p-i+1}(\mathbf{S}_w)} \min_{\mathbf{W}^*} \|\mathbf{W}^{k+1} - \mathbf{W}^*\|_F \\
&\leq \frac{2\sqrt{r}\lambda_1(\mathbf{S}_w)}{\sum_{i=1}^r \lambda_{p-i+1}(\mathbf{S}_w)} \sqrt{\frac{2\sum_{i=1}^r \lambda_i(\mathbf{S}_w)}{\lambda_r - \lambda_{r+1}}(F^* - F(\mathbf{W}^0))(1 - \frac{1}{\kappa(\mathbf{S}_w)})^{\frac{k+1}{2}}}.
\end{aligned} \quad (59)$$

We can see that when $k \to \infty$ in Eq. (59), we have $\frac{F^* - F(\mathbf{W}^{k+1})}{F^* - F(\mathbf{W}^k)} = 0$, which is the superlinear convergence rate, and we have

$$\frac{F^* - F(\mathbf{W}^k)}{F^* - F(\mathbf{W}^0)} \leq S^k(1 - \frac{1}{\kappa(\mathbf{S}_w)})^{\sum_{n=1}^k \frac{n}{2}} = S^k(1 - \frac{1}{\kappa(\mathbf{S}_w)})^{\frac{k(k+1)}{4}}, \quad (60)$$

where $S = \frac{2\sqrt{r}\lambda_1(\mathbf{S}_w)}{\sum_{i=1}^r \lambda_{p-i+1}(\mathbf{S}_w)}\sqrt{\frac{2\sum_{i=1}^r \lambda_i(\mathbf{S}_w)}{\lambda_r - \lambda_{r+1}}(F^* - F(\mathbf{W}^0))}$. □

# 7 Experiments

## 7.1 Toy Experiment

To test our approach, we conduct a toy experiment classifying dogs and cats. Training data deliberately include mismatched pairs, as seen in Figure 6(a). Despite mislabeled images, both the robust method and robust 2D method, designed for outlier robustness, correctly classify all images for $q = 1$, as shown in the lower part of Figure 6(a). To further illustrate the robust metric learning method, we also provide the details in Figure 2 example. Given the following sentences: 'He/She is happy/angry', 'He/She is very/quite happy/angry' and 'Happy/Angry', with a total number of 14. We manually give 20 correct side constraints, for example: 'He is happy' and 'She is angry' in different sets, while 'He is quite happy' and 'Happy' in the same. Also, there are some adversarial examples, where 5 links are mistakenly given, say 'She is happy' and 'He is quite angry' should be separated but they are put in the same set. Each sentence $\mathbf{x}$ is represented as a

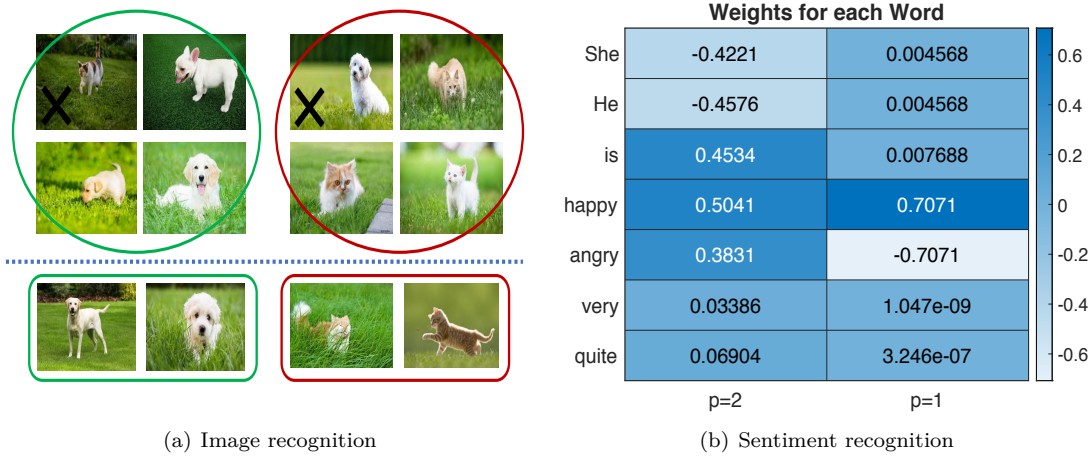

(a) Image recognition  (b) Sentiment recognition

Figure 6: Results from toy experiments. From left to right: (a) Image recognition: Training set (upper) and recognition results (lower). Even with the existence of incorrect pairs, our robust metric learning method still yields correct recognition; (b) Sentiment recognition: Our robust metric learning (p = 1) can learn reasonable weights for features even under adversarial attack while the traditional one (p =2) will fail.

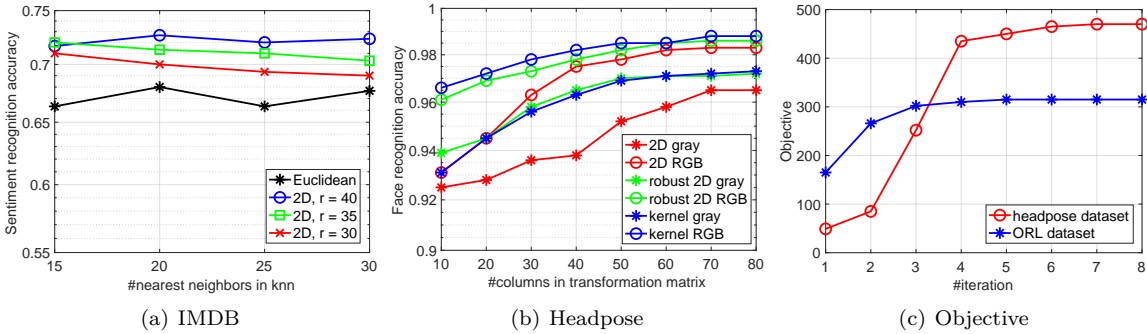

(a) IMDB  (b) Headpose  (c) Objective

Figure 7: From left to right: (a) Accuracy of the 2D metric learning with varying projected dimension r and $k$ in $k$-NN for the IMDB dataset; (b) Face recognition accuracy for the headpose dataset, where X-axis denotes the projected dimension; (c) The objective of the 2D metric learning is monotonically non-decreasing, same for the kernel method and the robust 2D method.

7-dimensional word-count vector (with each denoting a word such as 'happy', 'angry', *etc.*). As demonstrated in Figure 6(b), for vanilla squared Frobenius ($q = 2$), the learned weights for 'is', 'happy', and 'angry' are close, which is not as they should be. However, if we set $q = 1$, which is supposed to remain robust with outliers/attacks, it can obtain significantly different weights for 'happy' (0.707) and 'angry' (-0.707), while the rest are all set close to 0. This is in perfect accordance with our common sense, which indicates the potential application of our robust metric learning in other domains.

## 7.2 Sentiment Recognition

We evaluate the 2D metric learning method in a sentiment recognition task with the IMDB dataset. 5000 data instances are randomly sampled from the IMDB review dataset and are split into the training set and test set evenly. For each review, only the first 50 processed text words are embedded in a 50-dimension tensor and are labeled as positive or negative. We obtain **W** using samples from the training set, and then apply it when we use $k$-NN to do sentiment classification. We test the 2D metric learning method with varying numbers of columns in the transformation matrix and $k$ in the $k$-NN classifier. Figure 7(a) shows the result, our method can achieve stable and better performance than Euclidean distance-based classification.

Table 1: Time consumption (in seconds) and mean recognition accuracy (10 test runs) for headpose dataset

|  |  | MS | GMML | LMNN | KISSME | ITML |
|---|---|---|---|---|---|---|
|  | Time | 155.981 | 56.221 | 6601.881 | 792.331 | 5310.871 |
| Grayscale | $k = 3$ | 0.940 | 0.928 | 0.917 | 0.935 | 0.917 |
|  | $k = 5$ | 0.931 | 0.922 | 0.825 | 0.931 | 0.883 |
|  | $k = 9$ | 0.933 | 0.917 | 0.817 | 0.922 | 0.883 |
|  | Time | 312.212 | 93.125 | 9925.917 | 1238.569 | 7555.825 |
| RGB | $k = 3$ | 0.983 | 0.967 | 0.928 | 0.962 | 0.931 |
|  | $k = 5$ | 0.983 | 0.933 | 0.917 | 0.958 | 0.925 |
|  | $k = 9$ | 0.967 | 0.933 | 0.917 | 0.945 | 0.917 |
|  |  | MRL | RDML | **2D** | **Robust 2D** | **Kernel** |
|  | Time | 698.329 | 1005.72 | **2.062** | 8.185 | 2.517 |
| Grayscale | $k = 3$ | 0.933 | 0.933 | 0.933 | **0.967** | **0.967** |
|  | $k = 5$ | 0.917 | 0.933 | 0.933 | **0.967** | **0.967** |
|  | $k = 9$ | 0.883 | 0.917 | 0.933 | **0.967** | 0.933 |
|  | Time | 1025.978 | 2955.96 | 5.959 | 10.121 | **5.012** |
| RGB | $k = 3$ | 0.965 | 0.967 | **0.985** | **0.985** | **0.985** |
|  | $k = 5$ | 0.932 | 0.933 | 0.983 | **0.985** | 0.983 |
|  | $k = 9$ | 0.923 | 0.917 | 0.967 | **0.983** | **0.983** |

Table 2: Time consumption (in seconds) and mean recognition accuracy (10 test runs) for ORL dataset, with full test images, with random missing pixels as noise on test images, and with mismatched pairs

|  |  | MS | GMML | LMNN | KISSME | ITML |
|---|---|---|---|---|---|---|
|  | Time | 102.763 | 35.758 | 5721.998 | 603.992 | 4125.722 |
| Vanilla | $k = 3$ | 0.913 | 0.898 | 0.897 | 0.901 | 0.883 |
|  | $k = 5$ | 0.891 | 0.885 | 0.852 | 0.882 | 0.867 |
|  | $k = 7$ | 0.847 | 0.839 | 0.792 | 0.828 | 0.813 |
| Noise | $k = 3$ | 0.895 | 0.879 | 0.892 | 0.891 | 0.852 |
|  | $k = 5$ | 0.873 | 0.868 | 0.839 | 0.852 | 0.839 |
|  | $k = 7$ | 0.839 | 0.807 | 0.768 | 0.819 | 0.793 |
| Miss-matched | $k = 3$ | 0.892 | 0.872 | 0.853 | 0.858 | 0.863 |
|  | $k = 5$ | 0.875 | 0.827 | 0.822 | 0.839 | 0.817 |
|  | $k = 7$ | 0.819 | 0.808 | 0.767 | 0.795 | 0.792 |
|  |  | MRL | RDML | **2D** | **Robust 2D** | **Kernel** |
|  | Time | 465.827 | 805.386 | **2.287** | 8.552 | 2.817 |
| Vanilla | $k = 3$ | 0.905 | 0.902 | **0.933** | **0.933** | **0.933** |
|  | $k = 5$ | 0.887 | 0.872 | **0.917** | **0.917** | 0.902 |
|  | $k = 7$ | 0.835 | 0.851 | 0.861 | **0.867** | **0.867** |
| Noise | $k = 3$ | 0.857 | 0.867 | 0.892 | 0.913 | **0.917** |
|  | $k = 5$ | 0.825 | 0.845 | 0.873 | **0.892** | **0.892** |
|  | $k = 7$ | 0.817 | 0.769 | 0.833 | **0.877** | 0.837 |
| Miss-matched | $k = 3$ | 0.851 | 0.817 | 0.875 | **0.917** | 0.892 |
|  | $k = 5$ | 0.818 | 0.813 | 0.835 | **0.902** | 0.875 |
|  | $k = 7$ | 0.785 | 0.767 | 0.819 | **0.875** | 0.825 |

## 7.3 Image Segmentation and Recognition

We conduct image segmentation on some natural images (Martin et al., 2001) and evaluate our methods in an image classification task with $k$-NN classifier on two datasets, the headpose dataset (Gourier et al., 2004) and the ORL dataset (Samaria & Harter, 1994). For the headpose dataset, we conduct the experiment with two settings: **(1)** with grayscale images; **(2)** with RGB images. To deal with RGB images, we treat the three layers of images as three matrix blocks and append them together. Thus for an input image of size $p \times m$, in 2D methods we stretch it into a matrix of size $p \times 3m$. In other methods, we vectorize the input image to get

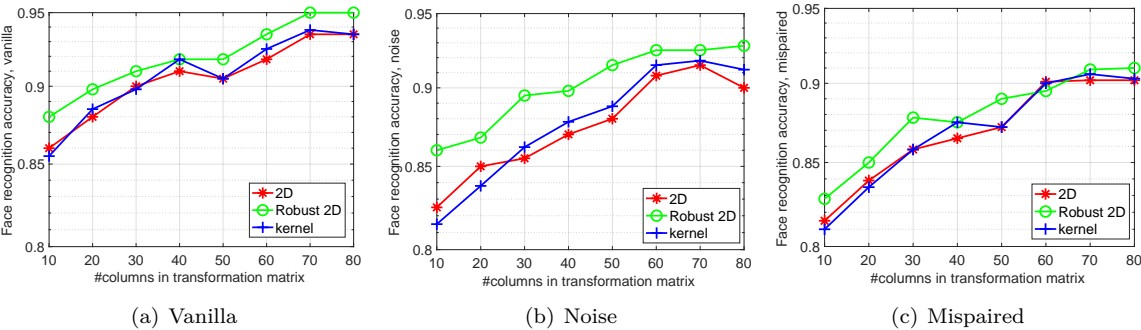

(a) Vanilla      (b) Noise      (c) Mispaired

Figure 8: Accuracy for the ORL dataset with varying projected dimensions, from left to right is obtained with vanilla images, with noise, and with mispaired images, respectively.

a vector with length $3 \times p \times m$. For the Robust metric learning, we set $q = 1$. To evaluate the robustness of the algorithms, we test on the ORL dataset in three situations: **(1)** learn with correct pairwise relationships, test on vanilla images; **(2)** learn with correct information but test on images with $10\%$ noises; **(3)** learn the metric while there exist $7\%$ miss-matched pairwise relationships, test on vanilla images. We also compare our method with several metric learning methods, including Multi-Similarity-based deep metric learning (MS) (Wang et al., 2019), GMML (Zadeh et al., 2016), ITML (Davis et al., 2007), LMNN (Weinberger et al., 2006), KISSME (Koestinger et al., 2012), MRL(-ADMM) (Lim et al., 2013), and RDML (Liu et al., 2019). Both GMML and MRL-ADMM claim their superiority in terms of low computation time, which will be compared with our methods in computation time. We evaluate both the time consumption and recognition accuracy of each algorithm. The time consumption to get $\mathbf{W}$, and the recognition accuracy with varying $k$ of $k$-NN classifier are listed in Table 1 and Table 2, best results are highlighted. It is obvious that the time consumption of our 2D methods and kernel method is a huge advantage and this merit is more significant with RGB images. Our methods require negligible computation time to get the transformation matrix, while other methods have much higher computational costs. Even compared with the GMML method, which serves as a breakthrough in terms of computation time, our proposed methods still have a great advantage in high efficiency. Our methods are able to achieve the best recognition accuracy almost on all tested datasets, at most times they are significantly better than most other methods, and they have comparable performance with deep metric learning on the two datasets. In Table 2, when we introduce noise and adversarial attack in the training process, the robust 2D metric learning method and robust metric learning method are proved to be able to achieve robust performance.

We also investigate the impact of varying the number of columns $r$ in the transformation matrix $\mathbf{W}$. By testing different values of $r$ while keeping $k$ of the $k$-NN classifier set to 3, Figure 7(b) illustrates that higher accuracy can be achieved as $r$ increases, reaching a threshold where the incremental gain diminishes. For a more detailed exploration of how the number of columns $r$ in the transformation matrix $\mathbf{W}$ influences accuracy, refer to Figure 8. Figure 7(c) illustrates the objective change in the 2D version, with kernel and robust methods exhibiting similar trends. Notably, an optimal transformation matrix $\mathbf{W}$ can be obtained within very few iterations. The presented results collectively affirm that the four proposed metric learning algorithms effectively address real-world cases. This validation aligns with our earlier analyses and suggests their superiority over their counterparts. Figure 9 showcases image segmentation results utilizing RGB pixel values and $X, Y$ locations as input features. The presence of red/green lines and pixels indicates must-link and cannot-link relationships, respectively. In the first row, images display user-specified pixels denoting the background and foreground. Subsequent rows reveal the impact of learning an appropriate distance metric transformation matrix $\mathbf{W}$ from user-specified pixels. The improved performance of the $k$-NN classifier in segmentation is evident. Our algorithm consistently outperforms GMML and RDML methods, as demonstrated in the lower rows. For instance, in the pyramid image, the black circled area, situated far from the labeled foreground region, is accurately classified by our method but not by others. Figure 10 demonstrates the results from the face recognition experiment. After imposing $10\%$ noise on the query image randomly, our method excels in identifying the nearest anchor images compared to others.

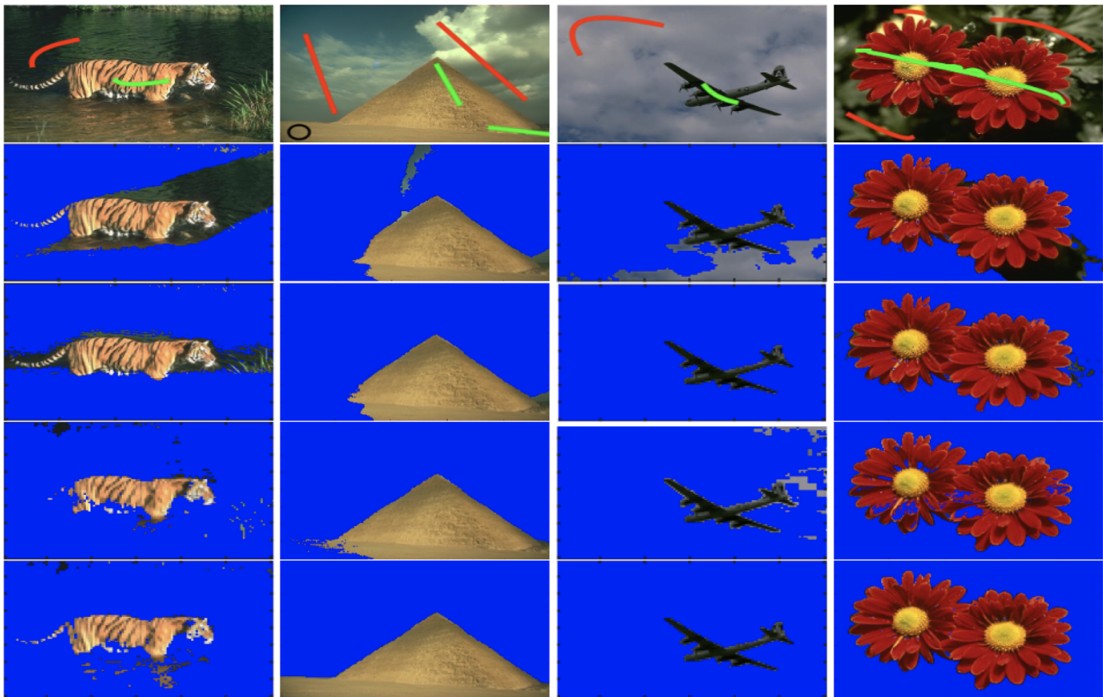

Figure 9: Image segmentation results. **First row:** Original images with labeled pixels. **Second:** Euclidean distance. **Third:** GMML method. **Fourth:** RDML method. **Fifth Row:** Robust Algorithm 2.

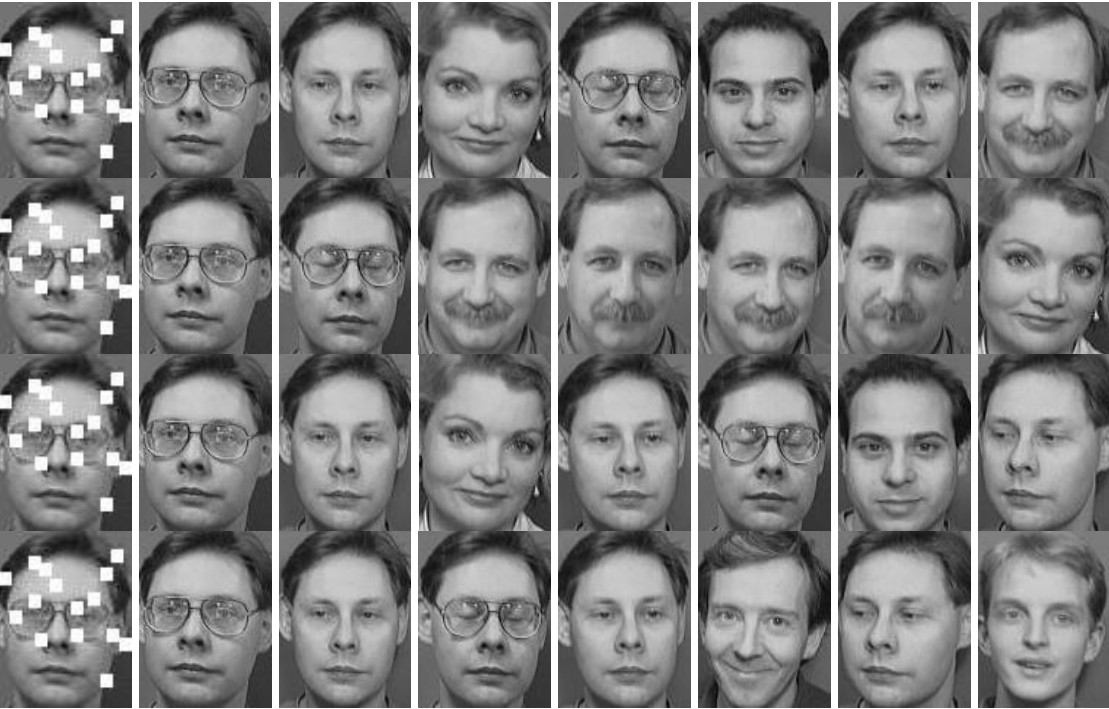

Figure 10: Face recognition results. **First row:** Euclidean distance. **Second row:** GMML method. **Third Row:** RDML method. **Fourth Row:** Robust Algorithm 4.

# 8   Conclusion

We introduce a robust metric learning approach that leverages the $\ell_{2,q}$-norm distance, offering enhanced resilience against data outliers and adversarial attacks. Additionally, we present 2D metric learning and kernel metric learning algorithms, tailored to mitigate the computational challenges associated with eigenvalue decomposition on high-dimensional covariance matrices. Our methods come with a strong theoretical foundation, ensuring the objective is monotonically increasing. Within each iteration, we obtain a closed-form optimal solution for 2D and kernel metric learning respectively. In addition, we rigorously establish the convergence rate of these proposed algorithms. Experiments on diverse real-world datasets are conducted to validate the effectiveness of our methods. The results highlight the efficiency and superior accuracy of our approaches in addressing a range of practical tasks in comparison with the counterparts.

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
