# OpenReview forum: "Robust Semi-Supervised Metric Learning Meets with High Dimensionality"
_TMLR — Rejected by TMLR_

### Review · Reviewer_Vcmd · 2024-04-30

**Summary Of Contributions:**

The paper focuses on metric learning problems. The authors proposed three new approaches such as the robust metric learning, 2D metric learning and kernelized metric learning. They presented some theories on the convergence issues and conducted some numerical experiments to verify the feasibility of the proposed algorithms.

**Audience:**

No

**Claims And Evidence:**

Yes

**Requested Changes:**

1.Please highlight the motivations of the algorithm and present the reason why the mentioned three algorithms are needed.
2.Figure 2 is actually meaningless.
3.Please highlight your contribution.
4.Please present a section to figure out the problem setting.
5.Please collect theoretical results in a unifed section and present more explanations on theoretical results and conditions.
6. in (2). Why such an M　is needed.
7. Two lines after (2), M is a metric and M is a matrix? What is M?

**Strengths And Weaknesses:**

In my own reading, the paper is not organized very well and is not in a good shape for publishing. It seems that the authors only list the algorithms and pile out the algorithms and results. There lacks motivation, the logical relation among algorithms and the detailed contribution descriptions. Futhermore, there are not any explanations and comparisons of theoretical results which seem to be the main contributions of the paper, since the numerical experiments are not sufficient. Due to this, I suggest to reject the paper and my main reason for the rejection is the motivation of proposing these three algorithms; why these algorithms are collected together in one paper with each one are not presented in a sufficient manner; the bad organization, and the somewhat trivial numerical results that cannot provide direct support of the proposed three approaches. The following are some other suggestions.

---

> ### Author Response · Authors · 2024-06-23
> **Response to Reviewer Vcmd**
>
> We are sorry that the paper's current organization makes the reviewer confused. We also would like to take this precious opportunity to clarify something and hope it helps:
> 1. why the mentioned three algorithms are needed
> A: Algorithm 2 solves the case where the data is not 2D, such as tabular data. Algorithm 3 is for 2D such as images and clean data with no corruption or adversarial examples. Algorithm 4 is a more generalized version (based on Alg 3) which can work for 2D data with missing value or incorrect labels. Algorithm 5 works for any order dimension data such as 3D, 4D. Besides, it can work when the original data is not linearly separable. It is similar to the case that we have vanilla PCA also we have kernel PCA.
>
> 2. Figure 2 is actually meaningless.
> A: Figure 2 wants to show the importance weight of each word can play a role for sentiment analysis. If we only count number of words, then 'He is very happy' would be very similar to 'He is very angry' as out of 4 words in each sentence, 3 are the same. But the sentiments of them are indeed different, as the importance weight of each word is different. Our goal is to learn the different weights for each word. Fig. 6b demonstrates that our proposed robust metric learning (here is a typo that it should be q=1 instead of p=1) can learn the weight well (only angry and happy matter with polar weights while the rest are close to 0) while vanilla metric learning (q=2) cannot.
>
> 3. Please highlight your contribution
> A: The items above Section 2 Related Work are our main contributions.
>
> 4. in (2). Why such an M is needed.
> A: Such M is to assign different weights for each feature. For example, assume x is a vector and M is a diagonal matrix, then x'Mx will put more weight on specific feature in x. If every feature they are of same importance then M is identity matrix, which is a special case that M is (semi) positive-definite. The goal is to learn the different weight, which is to find optimal M given some observations.
>
> 5. Two lines after (2), M is a metric and M is a matrix? What is M?
> A: M is a matrix, it is a metric for different features to assign different weights for different features. This indeed aligns with our common sense as well since not each feature is mutually important. For example, the height of a person is likely more important than the length of her/his hair to determine whether s/he is a good fit for basketball.

---

### Review · Reviewer_4BoN · 2024-05-01

**Summary Of Contributions:**

The paper proposed robust metric learning for data with related and unrelated pairs, throuch the use of $l_{2,p}$ norm instead of the Euclidean norm. The paper also proposes two related methods, one for direct 2-D (image data) metric learning, and one for kernelized robust metric learning with $l_{2,1}$ norm. Convergence analysis is provided to show that the proposed algorithms has superlinear convergence rates. Experiments are conducted to show the effectiveness and efficiency of the methods.

**Audience:**

Yes

**Claims And Evidence:**

Yes

**Requested Changes:**

See Weakness.

1. Refine the related works section with a thorough introduction.

2. Refine the experiment section. Please add more description about the data and the data/label generating process. I suggest including results on large datasets.

3. Add the complexity of all the algorithms (if not present), including the kernel method.

**Strengths And Weaknesses:**

Strengths: The paper is well-written and easy to follow.

The algorithm design is clear and well-justified.

The algorithms have rigorous theoretical guarantee on the convergence.

=========================================

Weakness:

1. The related work and introduction to previous methods are lacking. Please include more thorough introduction on metric learning, including, for example, the competing methods used in the experiments.

2. It seems that the kernelized version requires a lot computation and is hard to scale. There is no complexity analysis. Could you add one?

3. The experiments are not strong enough. The tested datasets are small. Please include the dataset description/stats in the paper, as well as more details on how the postive and negative pairs are generated.

---

> ### Author Response · Authors · 2024-06-23
> **Response to Reviewer 4BoN**
>
> We greatly appreciate the constructive feedbacks provided by our reviewer.
> 1. The related work and introduction to previous methods are lacking.
> A: Yes, we will add more methods appropriately, specifically, we will definitely add those to be compared in experiment section.
>
> 2. It seems that the kernelized version requires a lot computation and is hard to scale. There is no complexity analysis.
> A: The key part of kernel version is Eq. (41), which is to find generalized eigenvector \alpha given in Eq. (42). Apparently, the most computational cost is the inversion operation, which is O((s+d)^3), where s denotes the number of must-link constraint and d is the number of cannot-link constraint. Usually, in semi-supervised learning, s+d wouldn't be too large, or we can select partial of the information to make it relatively small. However, the key point here is even when the size of image is not large, for example 92*112, traditional method by vectorizing the image will be at least 10K complexity level. In short, kernel method will allow us to reduce the order of complexity if the dimension of data is large.
>
> 3. The experiments are not strong enough. The tested datasets are small. Please include the dataset description/stats in the paper, as well as more details on how the postive and negative pairs are generated.
> A: We will add more experiments on larger dataset such as CIFAR 100. For the experiments in the current paper, say ORL, we have the labels (40 subjects) of each image, positive pair means they are from the same subject (person) while negative means from different subjects. We simply generate by generating random number in Matlab.
>
> 4. Add the complexity of all the algorithms (if not present), including the kernel method.
> A: Great and thanks for the suggestion. We will add complexity analysis comparison in a table for later version.

---

### Review · Reviewer_qVN9 · 2024-06-09

**Summary Of Contributions:**

The authors propose a method to enhance the robustness in semi-supervised metric learning, especially for high-dimensional data. According to the authors, the contributions are three folds:

i) Motivated by the development of robust PCA, they also used the l2,q norm to enhance the learning objective in metric learning, such that the objective is less sensitive to the outliers.

ii) The authors proposed methods that are highly efficient in terms of computation.

iii) The authors further extend their proposed method into kernel version for handling more complex problems.

**Audience:**

Yes

**Claims And Evidence:**

Yes

**Requested Changes:**

There are a few issues the authors could potentially try to address:

On the experimental claims:
-- Though the major selling point is robustness, most of the analysis of this paper focus on i) improving the computational efficiency, and ii) the overall accuracy boost. The objective and subjective analysis in terms of robustness is not sufficient.

-- Though addressing high-dimensional data is another motivation of the proposed methods, the experimental results do show the promising trend; The authors mainly experiment with relatively small and low-dimensional datasets.

-- The kernel-based approaches do not seem to outperform non-kernel methods, like shown in Figure 8.


Besides the classic metric learning, deep metric learning has gained attention in recent decades. How’s the proposed methods compared to the deep metric learning metrics (e.g., Contrastive loss, Arcface loss) in terms of overall performance and robustness. Also, as deep metric learning is widely applied, will the proposed method also help with existing deep metric learning frameworks can be discussed.


Literature survey and related work discussion can be more adequate:

-- Though l2,q norm can help the learning objective to be less sensitive to learning objective; Sometimes, metric learning approach can also be used to do novelty/anomaly detection, like ‘Metric learning for novelty and anomaly detection by Marc Masana etc’ and ‘Hyperbolic Metric Learning for Visual outlier detection’. The authors could discuss the connection between the two directions.

-- There are also closely related works that are not discussed or cited, like ‘Algorithm Robustness for Semi-supervised metric learning’ by Marina-Irina et al.’ and ‘Double L2,p-norm based PCA for feature extraction’.

**Strengths And Weaknesses:**

Overall, given the success of robust PCA for feature extraction, one of the biggest motivations for this work to use l2,q norm also makes sense. The authors outlined algorithms for solving the objective under l2,q norm, and also providing the convergence analysis. The experiments compared with previous methods show the clear benefits of the proposed methods in terms of both accuracy in a few tasks and also the computation cost.

---

> ### Author Response · Authors · 2024-06-23
> **Response to Reviewer qVN9**
>
> We would like to thank the reviewer for detailed comments.
> 1. The authors mainly experiment with relatively small and low-dimensional datasets.
> A: If we are allowed to make revisions, we are going to conduct more experiments on larger dataset. Currently, the image, for example from ORL dataset is 92*112. Traditional method by vectorization will be very time consuming as Table 2 (Time row) demonstrates. If the size of image increases, it will be way longer to wait due to the eigenvalue decomposition operation involved.
>
> 2. The kernel-based approaches do not seem to outperform non-kernel methods, like shown in Figure 8.
> A: True, we didn't claim the kernel method will produce the very best performance, we provide it as a tool just like kernel PCA, kernel regression etc. The reason is we don't do fine-tuning to find the best kernel for the dataset which will be a different topic. We simply provide a framework for kernel version and different datasets probably have different kernel and parameters which will yield optimal results.
>
> 3. Also, as deep metric learning is widely applied, will the proposed method also help with existing deep metric learning frameworks can be discussed.
> A: Good suggestion! We will try to have discussions if we have opportunity (say the feedback is major/minor revision etc.). What flashes to my mind is the idea is somewhat similar (please refer to the footnotes on Page 2), in terms of setting the distance from different images/classes large while same images/class small. However, due to the sophisticated nature of deep learning such as nonconvexity, nonlinear layer etc, especially in our paper we have strong constraint (W'W=I), I guess the story wouldn't be the same.
>
> 4. Literature survey and related work discussion can be more adequate.
> A: Yes, we consider the paper is rather long (>20 pages), we left literature survey mainly to 'Brian Kulis et al. Metric learning: A survey. Foundations and Trends® in Machine Learning, 5(4):287–364, 2013.' as mentioned in the last sentence of paragraph 1. We really want to thank the reviewer to provide different perspectives, which we definitely will include and discuss in later version if possible.

---

### Decision · Action_Editor_ELYG · 2024-09-11

**Recommendation:** Reject

**Comment:**

The paper proposed a robust distance metric learning method using the $\ell_{2,q}$-norm.  While the proposed method is validated by some experiments,  the proposal is not supported by sufficient evidence.  For instance, the major selling point in this paper is the robustness. However, most of the analysis of this paper focus on improving the computational efficiency and the overall accuracy boost. The  analysis in terms of robustness is not sufficient since the experiment is only done on relatively small and low-dimensional datasets. In addition, besides the classic (linear) metric learning on which the paper focused, deep metric learning has gained attention in recent decades. The authors should also add more discussion on the work related to deep metric learning.

All three reviewers voted for rejection in their final recommendation. I concur with the reviewers and therefore cannot recommend its acceptance.

**Audience:**

metric learning for high dimensional data will be of interest to the machine learning community.

**Claims And Evidence:**

The proposed methods are not validated in the large-scale datasets and the current experiments are not sufficient to support the improvement of the proposed method over the existing ones.